# Structures of the human LONP1 protease reveal regulatory steps involved in protease activation

Mia Shin[1,2], Edmond R. Watson[1], Albert S. Song [1,2], Jeffrey T. Mindrebo[1], Scott J. Novick[3], Patrick R. Griffin [3], R. Luke Wiseman [2✉] & Gabriel C. Lander [1✉]

The human mitochondrial AAA+ protein LONP1 is a critical quality control protease involved in regulating diverse aspects of mitochondrial biology including proteostasis, electron transport chain activity, and mitochondrial transcription. As such, genetic or aging-associated imbalances in LONP1 activity are implicated in pathologic mitochondrial dysfunction associated with numerous human diseases. Despite this importance, the molecular basis for LONP1-dependent proteolytic activity remains poorly defined. Here, we solved cryo-electron microscopy structures of human LONP1 to reveal the underlying molecular mechanisms governing substrate proteolysis. We show that, like bacterial Lon, human LONP1 adopts both an open and closed spiral staircase orientation dictated by the presence of substrate and nucleotide. Unlike bacterial Lon, human LONP1 contains a second spiral staircase within its ATPase domain that engages substrate as it is translocated toward the proteolytic chamber. Intriguingly, and in contrast to its bacterial ortholog, substrate binding within the central ATPase channel of LONP1 alone is insufficient to induce the activated conformation of the protease domains. To successfully induce the active protease conformation in substrate-bound LONP1, substrate binding within the protease active site is necessary, which we demonstrate by adding bortezomib, a peptidomimetic active site inhibitor of LONP1. These results suggest LONP1 can decouple ATPase and protease activities depending on whether AAA+ or both AAA+ and protease domains bind substrate. Importantly, our structures provide a molecular framework to define the critical importance of LONP1 in regulating mitochondrial proteostasis in health and disease.

[1] Department of Integrative Structural and Computational Biology, Scripps Research, La Jolla, CA, USA. [2] Department of Molecular Medicine, Scripps Research, La Jolla, CA, USA. [3] Department of Molecular Medicine, Scripps Research, Jupiter, FL, USA. ✉email: wiseman@scripps.edu; glander@scripps.edu

Mitochondria are the site of essential cellular functions including oxidative phosphorylation, apoptotic signaling, calcium regulation, and iron-sulfur cluster biogenesis. Maintaining mitochondrial protein homeostasis (or proteostasis) is critical for mammalian cell viability[1,2]. To prevent the accumulation of misfolded, aggregated, or damaged proteins, mitochondria evolved a network of quality control proteases that function to regulate mitochondrial proteostasis in the presence and absence of cellular stimuli[1,3]. This includes the ATP-dependent protease LONP1—a protease that is conserved throughout evolution from bacteria to humans[1,3–5]. In mammals, LONP1 is a primary quality control protease responsible for regulating proteostasis and various biological functions within the mitochondrial matrix through diverse mechanisms. LONP1 proteolytic activity is involved in the degradation of misfolded and oxidatively damaged proteins, components of the electron transport chain, and mtDNA regulatory factors such as TFAM and POLG[4–10]. Moreover, LONP1 possesses ATP-dependent, protease-independent chaperone activity and works in concert with the mtHSP70-DNAJA3 chaperone system to solubilize a significant portion of imported mitochondrial proteins, including OXAL1, NDUFA9, and CLPX[11–13]. Homozygous deletion of *Lonp1* in mice is embryonically lethal, underscoring LONP1's importance in regulating mitochondrial function during organismal development[14]. Furthermore, imbalances in LONP1 activity are implicated in mitochondrial dysfunction associated with diverse pathologic conditions including organismal aging, cancer, and numerous other human diseases[1,4,14–17]. Mutations in *LONP1* are causal for cerebral, ocular, dental, auricular, skeletal (CODAS) syndrome, a multi-system developmental disorder consisting of a wide array of clinical manifestations including hypotonia, ptosis, motor delay, hearing loss, postnatal cataracts, and skeletal abnormalities[18–20]. Despite the central importance of LONP1 for regulating mitochondria in health and disease, the structural basis for human LONP1 proteolytic activation and activity remains poorly defined.

The LONP1 protease consists of an N-terminal domain involved in substrate recognition and oligomerization, a AAA+ (ATPases associated with diverse cellular activities) domain that powers substrate translocation, and a C-terminal serine protease domain involved in substrate proteolysis[1,4,21]. Numerous biophysical approaches have been employed to gain insight into the macromolecular structure and function of bacterial Lon[22–30], and recent cryo-electron microscopy (cryo-EM) studies of bacterial Lon in both substrate-free and substrate-bound conformations revealed the structural basis for substrate processing and protease activation[31]. However, few structural studies have been directed toward understanding the molecular mechanism of protease activation and substrate processing by human LONP1. A prior crystal structure of the isolated protease domain of human LONP1 (PDB:2X36) revealed the structural conservation between human and bacterial Lon protease domains, suggesting similar mechanisms of action[32]. Further, a low-resolution cryo-EM structure of human LONP1 showed the ATPase and protease domains organized to form a pseudo sixfold symmetric chamber, while the N-terminal domains are arranged as a trimer of dimers atop the AAA+ domains[33]. Given the complex proteolytic demands of the mitochondrial matrix, we aimed to define the structural and mechanistic differences from bacterial Lon that have evolved to improve the activity or regulation of this protease for mitochondrial regulation.

To address this deficiency, we solved cryo-EM structures of human LONP1 in the presence and absence of a translocating substrate. We show that in the absence of substrate, LONP1 adopts an open, left-handed spiral conformation that is bound to ADP and has its protease active sites organized in an inactive conformation. In the presence of substrate, the AAA+ domains of human LONP1 adopt a right-handed spiral configuration similar to that observed for many other AAA+ proteases[21,34,35]. While substrate binding in the ATPase channel of bacterial Lon was previously shown to be sufficient to induce the protease domains to organize into a sixfold symmetric ring with activated protease active sites[31], we were surprised to find that the protease domains in substrate-bound human LONP1 adopt an asymmetric arrangement with inactivated protease active sites. Treatment of LONP1 with the covalent inhibitor bortezomib induces a sixfold symmetric arrangement of the protease domains and renders all six protease domains in the active conformation, similar to that observed for bacterial Lon[31]. We hypothesize bortezomib serves as a substrate-like moiety that activates the protease domain. This peptidomimetic-induced rearrangement of the protease domains suggests that human LONP1 has evolved an additional level of regulation for protease activation, potentially as a means of tuning LONP1-dependent regulation of mitochondrial proteostasis and function.

## Results

**Substrate-bound and substrate-free structures of human LONP1.** Mature human LONP1 lacking its N-terminal mitochondrial targeting sequence was purified as stable, soluble hexamer using size-exclusion chromatography (Supplementary Fig. 1, see "Methods"). Purified LONP1 was incubated with saturating amounts of the slowly hydrolyzing ATP analog, ATPγS, a strategy commonly utilized to stabilize AAA+ proteins for structural studies[36–39]. Single-particle cryo-EM analyses resulted in reconstructions of two distinct conformations of the protease: one structure containing a peptide substrate trapped in its central channel and the other devoid of substrate (Fig. 1 and Supplementary Fig. 2). The presence of substrate in a subset of the complexes was attributed to either a co-purified endogenous protein substrate or self-degradation product and is consistent with the presence of endogenous substrate observed previously in cryo-EM studies of numerous other AAA+ proteins[21,34,35,37,40–43]. In both structures of human LONP1, residues ~420–947, comprising a portion of the N-terminal helical domain (NTD³ᴴ), the AAA+ domain consisting of the small and large ATPase subdomains, and the protease domain, were well-resolved (Fig. 1). Blurred EM density corresponding to the ~300-residue N-terminal domain was discernible in two-dimensional averages (Supplementary Fig. 3) but was not resolvable in the three-dimensional analyses, likely due to the inherent flexibility of this region.

The substrate-free human LONP1 reconstruction identified in our dataset was resolved to ~3.4 Å resolution, which was sufficient for atomic modeling of five of the six LONP1 protomers (Supplementary Fig. 2 and Supplementary Table 1). The organization is reminiscent of the left-handed, open lockwasher configurations previously observed for substrate-free bacterial Lon[29,31], although human LONP1 appears to oligomerize with a slightly smaller helical pitch than that of bacterial orthologues (Fig. 1a, b and Supplementary Fig. 4). The uppermost subunit of substrate-free human LONP1 hexamer demonstrates substantially more flexibility than its bacterial counterpart, as it is only discernible at low resolution in a subset of the particles (Supplementary Fig. 2). Due to the low resolution of this uppermost subunit, a copy of the middle protomer from the five-subunit spiral was rigid body fit into the low-pass filtered reconstruction for representational purposes. Similar to previous substrate-free structures of bacterial Lon[29,31], the subunits of human LONP1 in the absence of substrate are bound to ADP and all protease domains adopt an inactive conformation (Supplementary Figs. 4 and 5). The presence of a fully ADP-bound

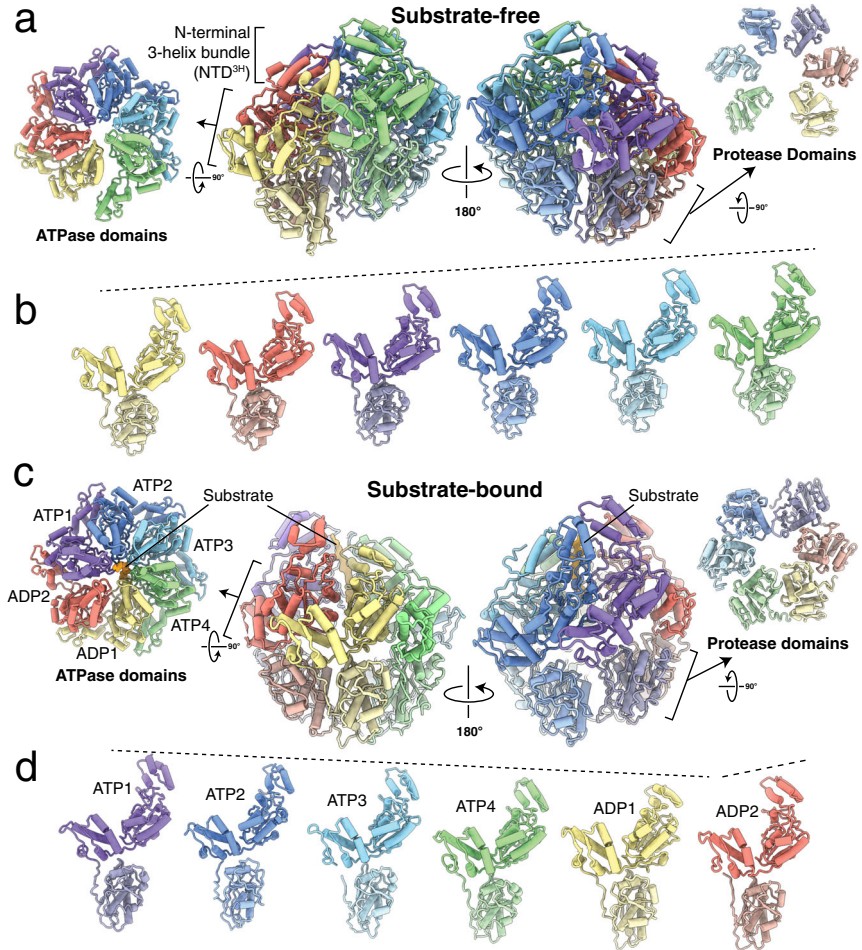

**Fig. 1 Architectures of substrate-free and substrate-bound human mitochondrial LONP1 protease. a** Lateral views of the substrate-free human LONP1 atomic model in a left-handed spiral configuration (center) flanked by axial exterior views of the ATPase (left) and protease (right) domain rings. Each subunit of the homohexamer is assigned a distinct color depending on its position in the spiral staircase, with the protease domains slightly desaturated in color relative to the ATPase domains. The same coloring is used throughout the figure. **b** The individual protomers of substrate-free human LONP1 lined up side-by-side, showing the ascending positions of the domains in the structure relative to the lowermost subunit. **c** Lateral views of the substrate-bound human LONP1 atomic model in a right-handed spiral configuration (center) flanked by axial exterior views of the ATPase (left) and protease (right) domain rings. The cryo-EM density of the substrate is shown as a solid isosurface colored orange. **d** The relative height of individual protomers within the substrate-bound conformer is shown by aligning all subunits to a common view, accentuated by dashed lines shown above the NTD[3H].

LONP1 conformation, despite incubating the sample with saturating amounts of ATPγS, likely stems from two factors: the LONP1 sample was purified without supplementing ATP, and the presence of contaminating ADP in the ATPγS stock. Since Lon has a 10–20-fold higher affinity for ADP over ATP[44], it is likely that LONP1 was fully ADP-bound prior to incubation with ATPγS. Since ADP exchange for ATP is promoted by substrate binding[45,46], adoption of the substrate-bound state would be limited by the amount of free substrate present in the purified sample.

This substrate-free, ADP-bound structure likely represents the LONP1 conformation that is competent for substrate engagement, analogous to the previously described substrate-free bacterial Lon complexes[29,31]. According to the previously proposed mechanism of substrate engagement, the topmost subunit of the left-handed spiral is the first to undergo nucleotide exchange upon interaction with substrate, followed by subsequent nucleotide and substrate engagement events that lead to a reorganization of the AAA+ domains within the hexamer into an active state that is competent for substrate translocation. The similarity of the substrate-free human and bacterial structures suggests conservation of this substrate engagement mechanism.

The substrate-bound LONP1 structure was resolved to ~3.2 Å resolution, which enabled atomic model building and refinement (Supplementary Fig. 2 and Supplementary Table 1). The structural conservation of the ATPase domains from bacteria to human is considerable, as the protomers of the human LONP1 are in a nearly identical organization as their counterpart in *Y. pestis* Lon, with RMSDs between 0.9 and 1.2 Å (Supplementary Fig. 6). The overall organization of these ATPase domains in the substrate-bound human LONP1 generally resembles that of previously determined substrate-bound AAA+ proteins, wherein the ATPases form a closed spiral staircase around a centrally positioned substrate peptide.

However, our substrate-bound human LONP1 exhibits several notable differences from the previously determined substrate-bound *Y. pestis* Lon. Whereas the AAA+ ring of bacterial Lon consists of four descending ATPase domains in its spiral staircase with two ascending 'seam' subunits (Supplementary Fig. 7), our substrate-bound structure of human LONP1 contains five descending ATPase domain subunits with a single 'seam' subunit transitioning between the highest and lowest subunits (Fig. 1c, d). To rule out the possibility that this unique configuration of LONP1 is a result of using ATPγS to slow ATP hydrolysis, as

opposed to incorporating a Walker B mutation in the *Y. pestis* study, we introduced a Walker B mutation into LONP1 (E591 to A591). We determined the structure of this Walker B LONP1 mutant in the presence of ATP, and at a resolution of ~4.8 Å, its conformation is indistinguishable from that of the substrate-bound WT LONP1 in the presence of ATPγS (Supplementary Fig. 8, Supplementary Fig. 9, and Supplementary Table 1). After extensive 3D classification, all particles belonging to Walker B LONP1 appear to be substrate-bound, despite not adding substrate during protein purification or cryo-EM sample preparation. This suggests that Walker B LONP1 constructs form more stable complexes with endogenous co-purified substrate or self-degradation products during purification and sample preparation than WT LONP1.

While the substrate-bound LONP1 structure contains features that distinguish it from the previously determined substrate-bound *Y. pestis* Lon, our LONP1 structure is generally consistent with substrate-bound structures of AAA+ protein translocases involved in a wide range of biological functions from bacteria to mammals[21,35,37,40–43]. Accordingly, the four topmost subunits of the staircase are bound to ATPγS (hereafter referred to as ATP for simplicity) while the lowest subunit contains ADP in its nucleotide-binding site (Fig. 1c, d and Supplementary Fig. 10). Density corresponding to an ADP molecule is also observed in the 'seam', or transitional subunit. To maintain consistency with other AAA+ proteins, we refer to the ATP-bound subunits as ATP1-4 (ATP1 as the topmost subunit), while the lowest subunit in the spiral staircase and 'seam' subunit are referred to as ADP1 and ADP2, respectively (Fig. 1d and Supplementary Fig. 10).

**Multiple pore loop interactions facilitate substrate transloca-tion in human LONP1.** The distinct configuration of the ATPase domains observed in human LONP1, relative to bacterial Lon, influences interactions with the engaged substrate. AAA+ proteins engage translocating substrates via pore loop residues within the ATPase domain that extend toward the central channel of the ATPase ring[21]. Previous structures of substrate-engaged AAA+ proteins have shown that aromatic residues in the ATPase pore loops intercalate into the backbone of an engaged peptide substrate every two amino acids to facilitate substrate translocation[21]. The ATPases in our human LONP1 reconstruction encircle a density accommodating a 12-residue peptide substrate (modeled as a polyalanine chain due to ambiguity of substrate sequence), which is surprisingly nearly twice the length of the engaged substrate visualized in substrate-bound *Y. pestis* Lon[31]. This observation provides an opportunity to examine the differences in substrate translocation by Lon proteases in bacteria and humans.

In human LONP1, the substrate-interacting Y565 in pore loop 1 from all four ATP-bound subunits (ATP1-4), as well as the ADP-bound subunit ADP1 at the bottom of the spiral staircase, are all observed engaging with the translocating substrate (Fig. 2a, b). This is in contrast to the substrate-interacting arrangement previously shown for bacterial Lon, where only the three uppermost ATP-bound subunits were tightly engaged with substrate, while the fourth subunit within the spiral weakly interacts with substrate[31]. Instead, the substrate interactions we observe in human LONP1 more closely resemble other substrate-bound structures of AAA+ proteins rather than bacterial Lon[21,34,35].

Notably, in addition to the pore loop 1 aromatic residue, we identified a second pore loop aromatic residue, Y599, within the AAA+ domains of human LONP1 that also engages the translocating substrate (Fig. 2a). This second pore loop residue assembles into a spiral that parallels the pore loop 1 aromatic, with Y599 from ATP1 and ATP2 engaged with the trapped substrate

(Fig. 2a). We refer to this second pore loop as pore loop 2. While Y599 from the other subunits do not appear to engage substrate in our structure of human LONP1, the spiraling organization of these residues and their proximity to translocating substrates suggest that these pore loop 2 aromatics may facilitate substrate guidance through the central channel and into the proteolytic chamber. Interestingly, the pore loop 2 aromatics of ATP1 and ATP2 increase the total number of interactions with the translocating polypeptide and are likely responsible for resolving more of the bound substrate as compared to our previous *Y. pestis* Lon structure. The presence of a methionine at the equivalent residue in *Y. pestis* Lon (M432) is likely responsible for a lack of observable pore loop 2 interactions in its structure, given that aromatic residues seem to be required for intercalating interactions with substrate (Fig. 2b and Supplementary Fig. 11)[31].

Consistent with an important role for the pore loop 2 residue Y599 in substrate translocation, recombinant LONP1 containing a Y599A mutation exhibited changes in both substrate proteolysis and ATP hydrolysis. LONP1[Y599A] showed a 66% decrease in the degradation rate of the model substrate FITC-casein (Fig. 2c), while showing a trend toward increased substrate-induced ATPase hydrolysis (Fig. 2d). This decreased degradation coupled with increased ATP consumption suggests that the Y599A mutant is a less efficient translocase compared to the wild-type enzyme, with increased likelihood of substrate "slippage" and unproductive ATP hydrolysis, as previously proposed for other AAA+ proteins[47]. Similar results were observed with the pore loop 1 mutant LONP1[Y565A], indicating that these two pore loop residues are both important for substrate processing. Other AAA+ translocases have been shown to utilize multiple pore loops to facilitate translocation, and it has been speculated that the integration of additional aromatic-containing pore loops into the translocation mechanism increases the ATPase motor's 'grip' on an incoming substrate[21,34–37,42]. Thus, inclusion of an aromatic residue, capable of intercalating into the backbone of an incoming substrate, into the pore loop 2 of human LONP1 may have evolved to increase interactions with substrate and, as a result, improve translocation and degradation efficiency.

**Substrate-bound human LONP1 maintains an inactive pro-tease conformation.** Another striking divergence of our substrate-bound structure of human LONP1 from bacterial Lon is in the organization of the protease domains. Previous results from bacterial Lon showed that substrate binding in four ATPase subunits and one ATP hydrolysis event promotes a rearrange-ment of the protease domains into a sixfold symmetric config-uration that releases the proteolytic active site from an inactive conformation, inducing the formation of a binding groove to position substrates for proteolysis[31]. Surprisingly, despite having a peptide substrate tightly bound within the channel, the pro-tease domains of substrate-bound human LONP1 do not adopt a sixfold symmetric configuration (Figs. 1c, 3a). Instead, the pro-tease domains of the substrate-bound hexamer follow the shal-low spiraling trajectory of the AAA+ domains, with an opening in the proteolytic ring between ATP1 and ATP2, the two uppermost subunits of the spiral staircase (Figs. 1d and 3a). Further, we observe all six protease active sites to be in an inactive configuration, wherein a loop containing the catalytic serine residue (S855) is folded into a $3_{10}$ helix, sterically occluding substrate access to the catalytic active site. Simulta-neously, an aspartic acid (D842) prevents the formation of the serine (S855)-lysine (K898) catalytic dyad, further hindering substrate degradation (Fig. 3b–e and Supplementary Fig. 12). These inactivated proteolytic active sites are consistent between WT LONP1 in the presence of ATPγS and Walker B LONP1 in

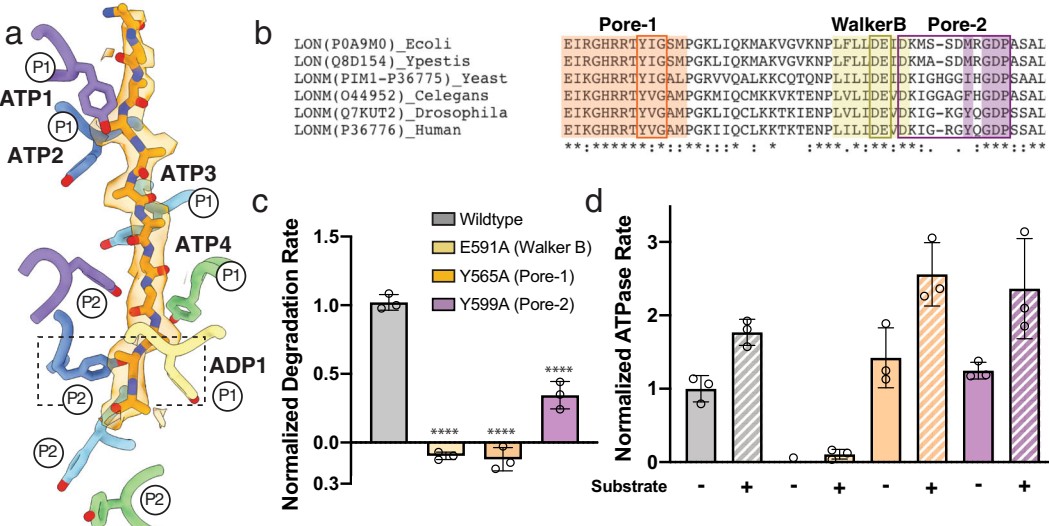

**Fig. 2 LONP1 employs two pore loop aromatic residues to facilitate substrate translocation. a** A twelve-residue polyalanine chain is modeled into the substrate density found in the substrate-bound human LONP1 structure, shown in a transparent orange surface representation. Y565 from pore-loop 1 (labeled P1) is shown using stick representations. Y565 residues from ATP1-4 and ADP1 subunits show intercalating, zipper-like interactions with substrate. Y599 from pore loop 2 (labeled P2) also shows intercalating interactions with substrate in the ATP1 (purple) and ATP2 (dark blue) subunits. **b** Sequence alignment between LONP1 homologs highlighting evolutionary conservation within pore loop 1 and Walker B motifs, as well as the evolutionary integration of a hydrophobic residue into pore loop 2 in metazoans. **c** Introducing pore loop mutations Y565A or Y599A shows a decrease in the degradation rate of a model substrate, FITC-casein. The hydrolysis-blocking Walker B mutation E591A is included as a control. Data are presented as bar graphs showing mean values with error bars showing standard deviation from three independent replicates. ****$p < 0.0001$ relative to wild type calculated by one-sided ANOVA. **d** Basal-level (filled bars) and casein-stimulated (hatched bar) ATPase rates are shown for the same mutations as in (**c**). Data are presented as bar graphs showing mean values with error bars showing standard deviation from three independent replicates. Error bars show standard deviation from three independent replicates.

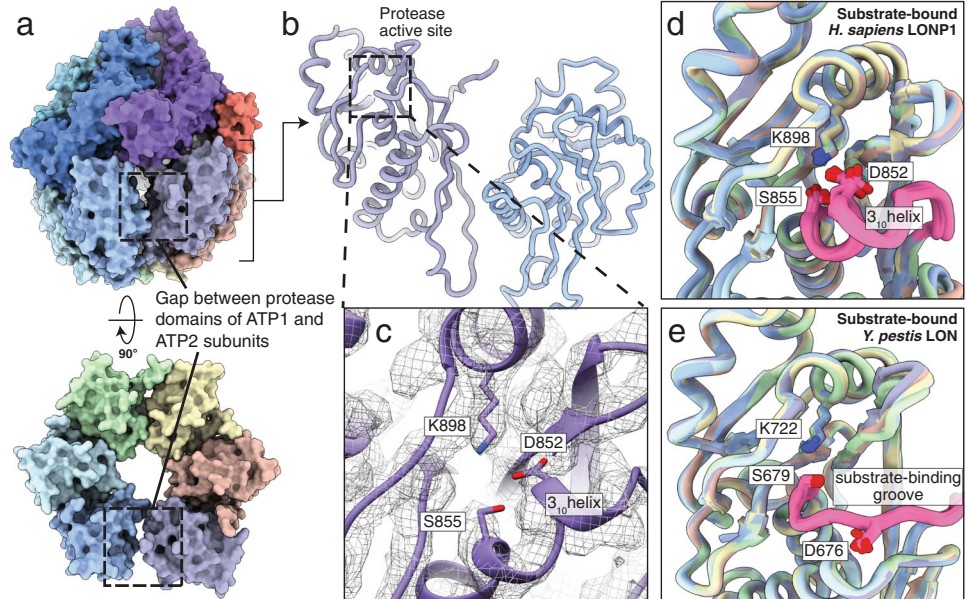

**Fig. 3 LONP1 in the substrate-bound conformation contains an inactive protease site. a** Surface representation of substrate-bound LONP1 showing a break in the protease domains between the highest subunit, ATP1, and its neighboring subunit, ATP2, highlighted using a dashed box. **b** The protease domains from ATP1 and ATP2 are shown using a ribbon representation, with the inactive active sites denoted by dashed boxes. **c** Close-up view of the proteolytic active site highlighting how the loop containing the catalytic serine (S855) is folded into a $3_{10}$ helix and an aspartic acid (D842) prevents the formation of the serine (S855)-lysine (K898) catalytic dyad. The electron microscopy density for this region is shown as a gray mesh. **d** Superimposing the six protease domains of the substrate-bound *H. sapiens* LONP1 shows that all protease domains adopt an inactive conformation, despite having substrate bound within the ATPase channel. **e** A superimposition of all six protease domains in substrate-bound *Y. pestis* LON shows that all protease domains adopt an activated conformation where the $3_{10}$ helix has unfolded to produce a binding groove to receive substrate, the aspartic acid (D676) is moved away from the active site, and the serine (S679)-lysine (722) catalytic dyad is positioned for peptide hydrolysis.

the presence of ATP (Supplementary Fig. 8). Further, this inhibited protease conformation is strikingly similar to those observed in the substrate-free structures of *Y. pestis* Lon[31] and human LONP1.

Previous results for bacterial Lon and other AAA+ proteases suggested that a flexible interdomain linker allows for large-scale conformational rearrangements of the AAA+ domains, independent of the symmetric protease, to facilitate substrate translocation[31,40,47]. It was proposed that when the protease domains were positioned laterally alongside one another underneath the translocation-competent ATPase ring, the catalytic serine-containing loop would extend toward the neighboring protease domain to stabilize inter-subunit interactions, resulting in the formation of the sixfold symmetric protease ring with six active proteolytic sites[31,48]. However, even though several of the protease domains in the substrate-bound human LONP1 structure position alongside one another in a similar fashion as those observed in the substrate-bound bacterial Lon structure, the catalytic loop remains in an inactive conformation (Fig. 3b–e, Supplementary Fig. 12). Whereas engagement of a substrate by the ATPase domains was itself sufficient for both symmetrizing and activating all six protease domains in bacterial LON, our structures show that this criterion is insufficient for protease activation of human LONP1. Instead, our substrate-bound structure of LONP1 indicates that there is an unmet regulatory requirement for activation of the protease domains. These findings underscore an evolutionary divergence in protease activation between bacterial and human LONP1, suggesting that other unidentified factors are involved in generating a proteolytically active human enzyme.

**Bortezomib induces the protease-active form in substrate-bound human LONP1.** We next sought to determine if we could induce a reorganization of the LONP1 proteolytic active site to the active conformation. We predicted that bortezomib, a peptidomimetic inhibitor of LONP1 that covalently engages the catalytic serine, could influence the protease conformation in LONP1[6,28]. Our prior cryo-EM structure of substrate-bound *Y. pestis* Lon showed that the conformation of the activated protease, even in the absence of bound peptide in the protease active site, closely resembled the crystal structure of a bacterial Lon protease bound to bortezomib[28]. This suggested that bortezomib could induce the activated protease conformation in LONP1. To test this prediction and characterize associated allosteric rearrangements, we determined a cryo-EM structure of substrate-bound human LONP1 incubated with saturating amounts of ATPγS and a 10-fold molar excess of bortezomib.

This complex, which we term LONP1[Bz] to indicate the addition of bortezomib, was resolved to ~3.2 Å resolution (Supplementary Fig. 13 and Supplementary Table 1). Density for the bortezomib molecules was clearly visible in each of the six active sites, covalently attached to the catalytic serine and positioned within the substrate-binding groove, mimicking a proteolytic substrate (Fig. 4a, d and Supplementary Fig. 15). As predicted, bortezomib association with the protease active sites led to their reorganization to the active form previously observed for bacterial Lon. Despite small differences in the spiraling ATPase conformations of substrate-bound human LONP1 and LONP1[Bz], both structures show a similar engagement of a 12-residue translocating peptide by pore loop 1 aromatics (Fig. 4b). Intriguingly, we were unable to determine the interactions between pore loop 2 aromatics and substrate in our LONP1[Bz] structure, as these loops were not as well-ordered as they were in the substrate-bound LONP1 structure, suggesting a dissociation of substrate from these pore loop 2 residues prior to or after substrate association with the protease domain.

Importantly, the protease domains in LONP1[Bz] switch from the asymmetric conformation observed in our substrate-bound LONP1 structures to adopt a sixfold symmetric organization (Fig. 4a). This suggests that the presence of substrate in the protease active site induces rearrangement of the serine-containing catalytic loop, such that it extends toward the neighboring protease domain to allow formation of the proteolytically active S855-K898 catalytic dyad. Previous reports suggest that this catalytic loop configuration is stabilized by the highly conserved residues V809, P854, and E884 from the adjacent protease domain (Fig. 4d and Supplementary Fig. 11)[28,31]. Consistent with an important role in protease activation, mutating these residues to alanine severely impairs human LONP1 proteolytic activity, while only minimally affecting ATP hydrolysis (Fig. 4e, f).

To confirm and further characterize the dynamics of protease domain configurations induced by bortezomib, we performed hydrogen–deuterium exchange mass spectrometry (HDX-MS) in the presence and absence of bortezomib (Fig. 4f and Supplementary Fig. 16). Consistent with our structures of substrate-bound LONP1 and LONP1[Bz], HDX-MS shows that the primary effect of bortezomib is to induce conformational remodeling of the protease domain. Notably, peptides comprising residues within the two helices flanking the catalytic S855-containing loop and the loop itself (peptides 803–820, 836–859, and 885–909) show reduced $D_2O$ exchange, thereby exhibiting significant stabilization upon addition of bortezomib (Fig. 4g and Supplementary Fig. 16). These results underscore the importance of these residues in maintaining the active form of the proteolytic active site. Importantly, these results, along with our structures of substrate-bound LONP1 and LONP1[Bz], support a model whereby engagement of substrate at the protease active site promotes conformational remodeling that includes inter-subunit stabilization of the catalytic loop region to promote proteolytic activation of human LONP1, distinguishing the protease activation mechanisms of bacterial and human LON homologs.

## Discussion

Here, we determine structures for human LONP1 in three different states: substrate-free, substrate-bound, and both substrate- and bortezomib-bound. These structures allowed us to establish a molecular framework to define the proteolytic activation of human LONP1. Surprisingly, our structures identify key differences between human and bacterial Lon despite high sequence conservation, which likely reflect evolutionary adaptations important for mitochondrial regulation. We show that human LONP1 integrates a second pore loop residue that allows increased interactions with the translocating peptide. These increased interactions likely provide the human protease enhanced 'grip' on substrates, potentially improving its ability to degrade damaged or non-native proteins within the mitochondrial environment, such as oxidatively-modified proteins previously shown to be LONP1 substrates[10,15].

Further, we show that the allostery associated with protease activation in human LONP1 is distinct from its bacterial homologs. Whereas substrate engagement by the AAA+ domains in *Y. pestis* Lon concomitantly promotes the displacement of a $3_{10}$ helix occluding the protease active site to drive organization of the serine/lysine catalytic dyad required for proteolytic activity, substrate binding within the ATPase of human LONP1 does not trigger rearrangement of the protease active sites. Rather, adoption of the activated protease within the substrate-bound conformation can be induced by the addition of the peptidomimetic LONP1 inhibitor bortezomib, which binds within the substrate-binding groove of the LONP1 active site. This suggests that

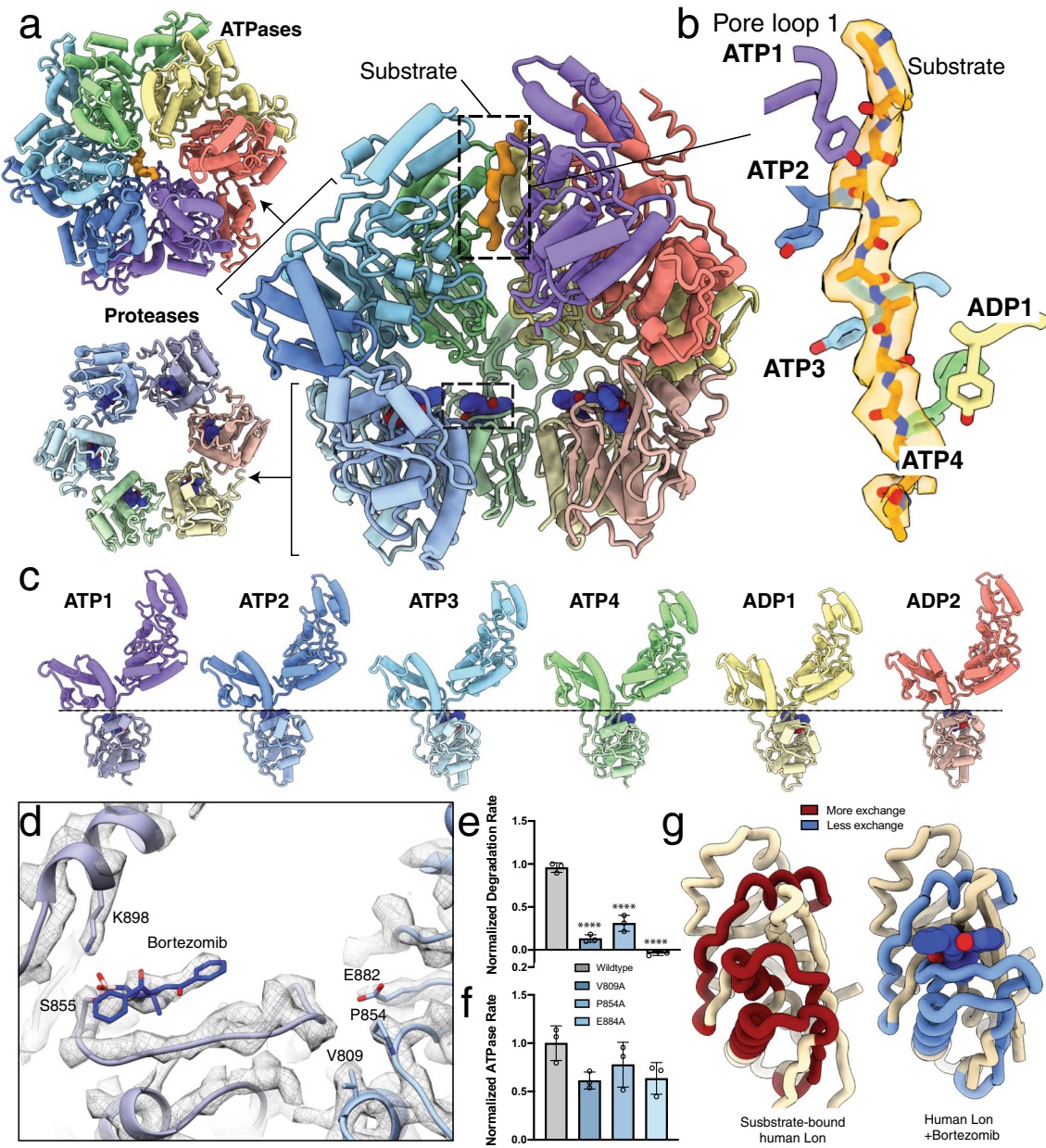

**Fig. 4 Bortezomib induces a sixfold symmetrization of the LONP1 protease domains and promotes activation of the proteolytic active site. a** Two external axial views of the bortezomib- and substrate-bound human LONP1$^{Bz}$ atomic model (left) is shown next to a cutaway lateral view of LONP1, using the same representation and coloring as in Fig. 1. Bortezomib (blue) bound within the protease active sites are shown using a sphere representation, with one bortezomib molecule denoted by a dashed box. **b** A twelve-residue polyalanine chain is modeled into the substrate density within the central channel of the LONP1$^{Bz}$ structure, shown as in Fig. 2. The pore loop 1 Y565 residues from subunits ATP1-4 maintain intercalating, zipper-like interactions with substrate; however, Y565 from ADP1 in LONP1$^{Bz}$ is positioned further from substrate as compared to our substrate-bound structure (see Fig. 2). **c** When the subunits are positioned side-by-side, the protease domains are aligned along a single plane. **d** Bortezomib covalently attached to S855 in ATP1 is shown using a ribbon representation within a mesh representation of the cryo-EM map, where the loop containing the catalytic serine S855 is observed extended toward the neighboring protease domain and interacting with conserved residues V809, P854, and E882 from the neighboring protease domain, shown in blue. **e, f** Graphs showing rates of FITC-casein degradation (**e**) or ATPase rate (**f**) for LONP1 harboring alanine mutations in the conserved residues V809, P854, and E882 involved in stabilizing the catalytic loop in the active conformation of the protease active site. Data are presented as bar graphs showing mean values with error bars showing standard deviation from three independent replicates. ****$p < 0.0001$ relative to wild type calculated by one-sided ANOVA. **g** A single protease domain is shown as a worm representation from the substrate-bound LONP1 structure in the absence (left) and presence (right) of bortezomib. Regions showing a decrease in $D_2O$ exchange measured by HDX-MS upon bortezomib binding are colored red in the inactive state (signifying greater exchange) and blue in the bortezomib-bound state (signifying decreased exchange).

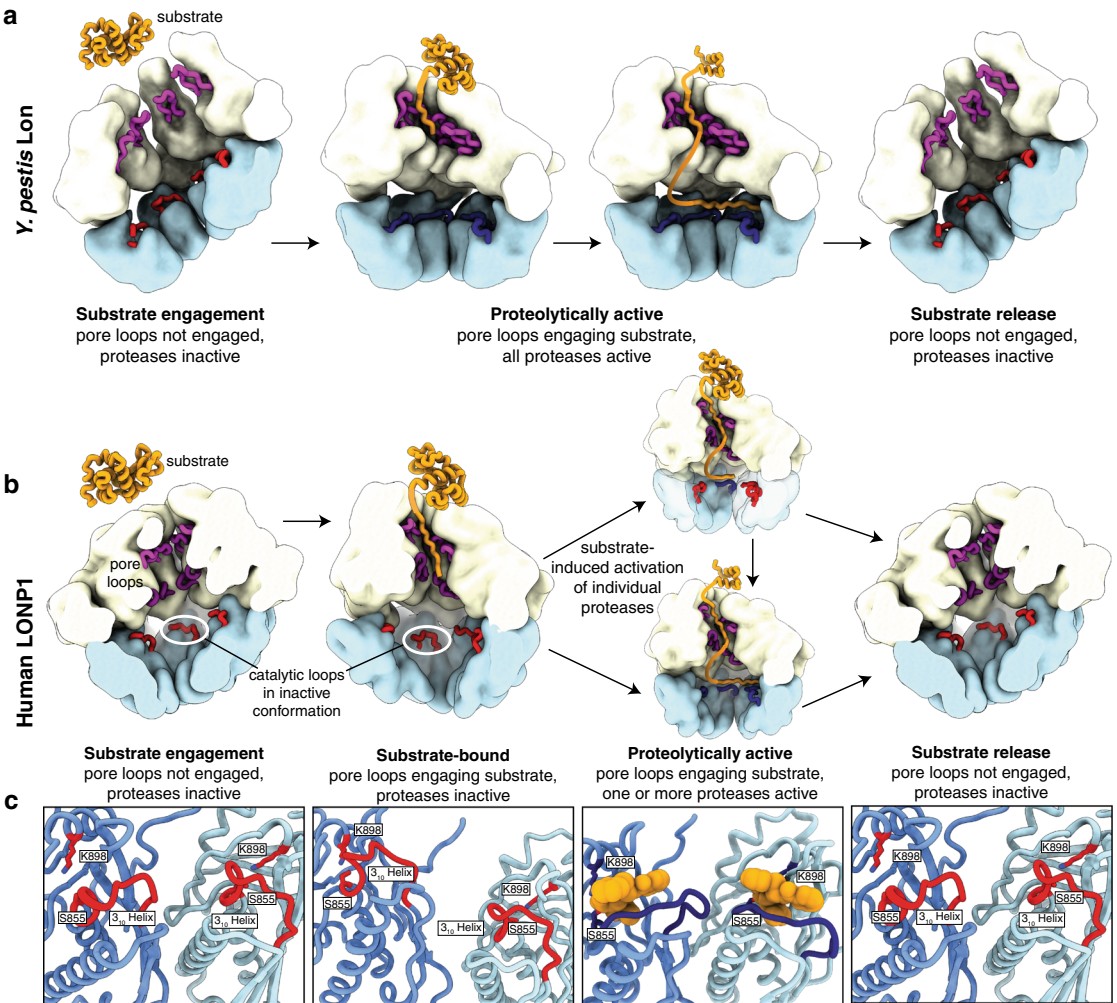

**Fig. 5 Models for protease activation in *Y. pestis* and human LONP1. a** In *Y. pestis* Lon, substrate is engaged by the open state conformer (left) with inactivated protease domains (red), causing a transition to a translocation-competent organization of the ATPases (middle, left). Substrate engagement by the pore loops (purple) in the central channel induces protease activation (dark blue). A conformational change to the activated, C6-symmetric protease conformation precedes substrate entering the protease active site (middle, right). Substrate is released upon Lon protease transitioning back to the open state conformer (right). **b** In Human LONP1, substrate is engaged by the open state conformer (left), causing a transition to a translocation-competent organization of the ATPases (middle, left). Despite substrate engagement by the pore loops (purple) in the central channel, the protease active sites remain in an inactive conformation (red). The protease domain is only activated in the presence of substrate (middle, right) in one (top) or more (bottom) protease domains, at which point the catalytic loop (red) extends away from the catalytic residues to enable substrate cleavage (dark blue). **c** Close-up views of the proteolytic active sites of substrate-free, substrate-bound, proteolytically active, and substrate-released LONP1 enzymes. In the inactive substrate-free and inactive substrate-bound states, a 3₁₀ helix sterically hinders substrates from entering the proteolytic active site for degradation in addition to preventing the formation of the catalytic dyad between S855 and K898. However, in the presence of substrate (represented by bortezomib in orange in the right panel) in the proteolytic active site, the catalytic serine-containing loop extends toward the neighboring subunit, the catalytic dyad is formed, and the protease domains activate for proteolytic cleavage.

substrate engagement with the protease active site (to displace the occluding 3₁₀ helix) is tightly linked to LONP1 proteolytic activity.

Although our structure of LONP1^Bz had bortezomib molecules bound to each of the subunits, thereby symmetrizing the protease domain, it is possible that all six protease active sites of human LONP1 are rarely, if ever, simultaneously activated for proteolytic activity. Rather, LONP1 could activate a few protease subunits at a time in order to cleave its substrates. Such a system is compatible with a processive degradation mechanism, as proteolytic peptides would be translocated toward the protease domains by the ATPase, and progressively cleaved as they encounter and activate individual protease active sites. Notably, this protease activation mechanism would enable human LONP1 to be specifically tuned to the types and quantities of substrates present

within the matrix, which continuously fluctuate in response to cellular queues. This fine-tuning of protease activity also presents a mechanism by which LONP1 could perform site-specific cleavage of substrates[49]. Further, the additional level of regulation of proteolytic activity for LONP1 revealed by our structures provides a mechanism to prevent the aberrant degradation of proteins important for mitochondrial function (Fig. 5).

Intriguingly, the ability to differentially regulate its ATPase and proteolytic activities through substrate engagement with distinct domains endows LONP1 with the functional plasticity necessary to regulate the diverse roles of this mitochondrial enzyme. Apart from its proteolytic activity, LONP1 also functions as an ATP-dependent chaperone within mitochondria, facilitating the folding of substrates including OXAL1 and NDUFA9[11]. Interestingly, protease inactive LONP1 retains chaperone activity, suggesting

that chaperoning and proteolytic degradation are distinct, independently regulated functions of this protein[11–13]. Our results indicate that LONP1 can decouple its ATPase and proteolytic activities through the differential engagement of substrates with the AAA+ and protease domains. This suggests that chaperone substrates may have evolved to interact with substrate-bound, proteolytically inactive LONP1, bypassing the allostery associated with protease activation and allowing effective ATP-dependent chaperoning independent of proteolytic activity. While additional studies are required to further pursue this potential mechanism, our results establish a mechanistic framework to define how proteolytic activity and ATPase-dependent unfolding can be differentially regulated, promoting diverse LONP1-dependent remodeling of proteins in mammalian mitochondria.

Last, our structures provide a structural basis for defining the pathologic and potentially therapeutic implications of LONP1 in human disease. For example, mapping known CODAS syndrome-associated mutations of human *LONP1* onto our structure of proteolytically active LONP1[Bz] highlights their position at interfaces between ATPase domains (Supplementary Fig. 17). This suggests that these mutations likely impact the tight subunit coordination required for ATP hydrolysis and substrate translocation. Consistent with this, CODAS mutations were previously shown to reduce LONP1 ATPase and proteolytic activities[20]. Given that the diversity of CODAS mutations, as well as the unique combination of mutations present in heterozygous individuals, result in a wide array of phenotypes in affected individuals, our atomic models of human LONP1 provide the structural basis for future analyses of CODAS patient mutations required to understand the diverse etiology of this disease. Further, our structures provide a molecular framework to facilitate the development of small molecule strategies to modulate LONP1 activity to mitigate pathologic mitochondrial dysfunctions associated with imbalances in LONP1 proteolysis[17,50].

## Methods

**Protein expression and purification.** A gene encoding an N-terminally His6-tagged human LONP1 with methionine 115 (the N-terminal residue of mature mitochondrial-matrix localized LONP1) as its first residue was inserted into a pET20b *E. coli* expression vector. The pET20b-*His6-LONP1* plasmid was transformed into the Rosetta 2(DE3)pLysS *E. coli* strain for recombinant protein expression. Cells were grown in 3% tryptone, 2% yeast extract, and 1% MOPS pH 7.2 with 100 μg/ml ampicillin and 25 μg/ml chloramphenicol, and cultured at 37 °C shaking at 220 rpm to an optical density (OD600) of 0.8. Protein overexpression was induced with 1 mM IPTG for 16 h at 16 °C. Cells were harvested by centrifugation at 5000 × g, resuspended in buffer A (0.2 M NaCl, 25 mM Tris pH 7.5, 20% glycerol, and 2 mM β-mercaptoethanol), flash-frozen and then stored at −80 °C for later use. Upon thawing, cells were lysed by sonication. Cell lysate was cleared by centrifugation at 27,143.1 × g and then added to TALON Cobalt resin equilibrated in Buffer A and loaded onto a gravity-flow column to allow His6-LONP1 binding. The column was washed with buffer A containing 20 mM imidazole to remove unbound proteins. Bound proteins were eluted in 1.2 column volumes of buffer A containing 300 mM imidazole. Eluted proteins were concentrated and loaded onto a Superose 6 size exclusion column equilibrated with buffer A. Fractions containing His6-LONP1 were pooled, concentrated, and flash-frozen for storage at −80 °C. The pET20b-*His6-LONP1* plasmid served as a template for generating *LONP1* mutants using the QuikChange™ site-directed mutagenesis approach (Supplementary Table. 2). Each LONP1 mutant protein was expressed and purified as described above.

**In vitro proteolysis assay.** Purified human LONP1 (0.1 μM) was incubated for 15 min at 37 °C in LONP1 activity buffer (50 mM Tris-HCl pH 8, 100 mM KCl, 10 mM MgCl2, 1 mM DTT, and 10% Glycerol) in the presence or absence of 2.5 mM ATP. FITC-Casein (0.8 μM; Sigma-Aldrich) was added to initiate the degradation reaction. An increase of fluorescence (excitation 485 nm, emission 535 nm) resulting from free FITC molecules was monitored using a TECAN plate reader. Three biological replicates were performed for each LONP1 mutant and the data were fit to a straight line from which the slope was extracted to calculate the rate of substrate degradation. GraphPad Prism software was used for data analysis. Mean and standard error of mean (SEM) were calculated by performing column statistics. ANOVA was used to calculate *p*-value, as indicated.

**In vitro ATP-hydrolysis assay.** In vitro ATP hydrolysis was carried out in LONP1 activity buffer (50 mM Tris-HCl pH 8, 100 mM KCl, 10 mM MgCl2, 10% Glycerol, and 1 mM DTT) in the presence or absence of 2.5 mM ATP. Casein (9.7 μM) was added to the reaction mixture when measuring substrate-induced ATPase activity. Reaction components and purified LONP1 (0.1 μM) were incubated separately at 37 °C for 5 min. LONP1 was added to initiate ATP hydrolysis and the amount of free inorganic phosphate at each time point was measured by adding Malachite Green Working Reagent (Sigma-Aldrich) and incubating components for 30 min at room temperature for color development before measuring absorbance at 620 nm (OD620). Three biological replicates were performed for each LONP1 mutant in the presence and absence of substrate. Data were fit to a straight line and the slopes were extracted to calculate ATP hydrolysis rates. GraphPad Prism software was used for data analysis. Mean and standard error of mean (SEM) were calculated by performing column statistics. ANOVA was used to calculate *p*-value, as indicated.

**Hydrogen–deuterium exchange (HDX) detected by mass spectrometry (MS).** Differential HDX-MS experiments were conducted as previously described with a few modifications[51,52]. Peptides were identified using tandem MS (MS/MS) with an Orbitrap mass spectrometer (Q Exactive, ThermoFisher). Product ion spectra were acquired in data-dependent mode with the top five most abundant ions selected for the product ion analysis per scan event. The MS/MS data files were submitted to Mascot (Matrix Science) for peptide identification. Peptides included in the HDX analysis peptide set had a MASCOT score >20 and the MS/MS spectra were verified by manual inspection. The MASCOT search was repeated against a decoy (reverse) sequence and ambiguous identifications were ruled out and not included in the HDX peptide set.

For HDX-MS analysis, LONP1 (10 μM) was incubated with 1 mM TCEP and 1 mM ATPγS for 5 min on ice and then Bortezomib (100 μM) was added and incubated at 37 °C for 30 min. Next, 5 μl of sample was diluted into 20 μl D2O buffer (50 mM Tris-HCl, pH 8; 75 mM KCl; 10 mM MgCl2) and incubated for various time points (0, 10, 60, 300, and 900 s) at 4 °C. The deuterium exchange was then slowed by mixing with 25 μl of cold (4 °C) 0.1 M sodium phosphate, 50 mM TCEP. Quenched samples were immediately injected into the HDX platform. Upon injection, samples were passed through an immobilized pepsin column (2 mm × 2 cm) at 200 μl min⁻¹ and the digested peptides were captured on a 2 mm × 1 cm C8 trap column (Agilent) and desalted. Peptides were separated across a 2.1 mm × 5 cm C18 column (1.9 μl Hypersil Gold, ThermoFisher) with a linear gradient of 4–40% CH3CN and 0.3% formic acid, over 5 min. Sample handling, protein digestion, and peptide separation were conducted at 4 °C.

Mass spectrometric data were acquired using an Orbitrap mass spectrometer (Exactive, ThermoFisher). HDX analyses were performed in triplicate, with single preparations of each protein–ligand complex. The intensity weighted mean *m/z* centroid value of each peptide envelope was calculated and subsequently converted into a percentage of deuterium incorporation. This is accomplished determining the observed averages of the undeuterated and fully deuterated spectra and using the conventional formula described elsewhere[52]. Statistical significance for the differential HDX data is determined by an unpaired *t*-test for each time point, a procedure that is integrated into the HDX Workbench software[53]. Corrections for back-exchange were made on the basis of an estimated 70% deuterium recovery, and accounting for the known 80% deuterium content of the deuterium exchange buffer.

For data rendering, HDX data from all overlapping peptides were consolidated to individual amino acid values using a residue averaging approach. Briefly, for each residue, the deuterium incorporation values and peptide lengths from all overlapping peptides were assembled. A weighting function was applied in which shorter peptides were weighted more heavily and longer peptides were weighted less. Each of the weighted deuterium incorporation values was then averaged to produce a single value for each amino acid. The initial two residues of each peptide, as well as prolines, were omitted from the calculations. This approach is similar to that previously described[54]. HDX analyses were performed in triplicate, with single preparations of each purified protein/complex. Statistical significance for the differential HDX data is determined by *t* test for each time point, and is integrated into the HDX Workbench software[53].

**Sample preparation for electron microscopy.** Wild-type human LONP1 was diluted to a concentration of 2.5 mg/ml in 50 mM Tris pH 8, 75 mM KCl, 10 mM MgCl2, 1 mM TCEP, and 1 mM ATPγS (Jena Bioscience, purity ≥90% (HPLC), contains <10% ADP). Samples were mixed and incubated on ice for 5 min to ensure nucleotide binding. Four microliters of the sample were applied onto 300 mesh R1.2/1.3 UltrAuFoil Holey Gold Films (Quantifoil) that were plasma cleaned prior to sample application for 7 s using a Solarus plasma cleaner (Gatan, Inc.) with a 75% nitrogen, 25% oxygen atmosphere at 15 W. Excess sample was blotted away for 4 s using Whatman No. 1 filter paper and vitrified by plunge freezing into a liquid ethane slurry cooled by liquid nitrogen using a manual plunger in a 4 °C cold room whose humidity was raised to 95% using a humidifier. For WalkerB mutant LONP1, LONP1 was diluted to a concentration of 2.5 mg/ml in 50 mM Tris pH 8, 75 mM KCl, 10 mM MgCl2, 1 mM TCEP, and 1 mM ATP. For bortezomib-bound LONP1, LONP1 was diluted to a concentration of 2.5 mg/ml in 50 mM Tris pH 8, 75 mM KCl, 10 mM MgCl2, 1 mM TCEP, and 1 mM ATPγS with the addition of 10-fold molar excess bortezomib. Samples for WalkerB and bortezomib-bound

LONP1 were prepared for cryo-EM analyses using the same procedures used for the wild-type sample.

**Electron microscopy data acquisition.** For wild-type human LONP1, cryo-EM data were collected on a Thermo-Fisher Talos Arctica transmission electron microscope operating at 200 keV using parallel illumination conditions[55]. Micrographs were acquired using a Gatan K2 Summit direct electron detector, operated in electron counting mode applying a total electron exposure of 50 e⁻/Å² as a 114-frame dose-fractionated movie during a 11.4-s exposure (Supplementary Fig. 2a). The Leginon data collection software[56] was used to collect 2912 micrographs at ×36,000 nominal magnification (1.15 Å/pixel at the specimen level) with a nominal defocus set to −1.5 μm. Variation from the nominally set defocus due to a ~5% tilt in the stage gave rise to a defocus range (this tilt was not intentional and required service to correct). Stage movement was used to target the center of sixteen 1.2-μm holes for focusing, and an image shift was used to acquire high magnification images in the center of each of the 16 targeted holes.

Similarly for WalkerB mutant LONP1, cryo-EM data were collected on a Thermo-Fisher Talos Arctica transmission electron microscope operating at 200 keV using parallel illumination conditions[55]. Micrographs were acquired using a Gatan K2 Summit direct electron detector, operated in electron counting mode applying a total electron exposure of 50 e⁻/Å² as a 59-frame dose-fractionated movie during a 11.8-s exposure (Supplementary Fig. 8a). The Leginon data collection software[56] was used to collect 2415 micrographs at ×36,000 nominal magnification (1.15 Å/pixel at the specimen level) with a nominal defocus set to −1.2 μm (a single value due to the tilted stage). Stage movement was used to target the center of four 1.2-μm holes for focusing, and an image shift was used to acquire high magnification images in the center of each of the four targeted holes.

Cryo-EM data for bortezomib-bound LONP1 were likewise collected on a Thermo-Fisher Talos Arctica transmission electron microscope operating at 200 keV using parallel illumination conditions[55]. Micrographs were acquired using a Gatan K2 Summit direct electron detector, operated in electron counting mode applying a total electron exposure of 50 e⁻/Å² as a 52-frame dose-fractionated movie during a 10.4-s exposure (Supplementary Fig. 12a). The Leginon data collection software[56] was used to collect 4776 micrographs at ×36,000 nominal magnification (1.15 Å/pixel at the specimen level) with a nominal defocus of −1.2 μm (a single value due to the tilted stage). Images for Bortezomib-bound LONP1 were collected using a similar strategy for wild-type LONP1. Micrograph frames for all three datasets were aligned using MotionCor2[57] and CTF parameters were estimated with CTFFind4[58] in real-time during data acquisition to monitor image quality using the Appion image processing environment[59].

**Image processing for wild-type human LONP1 + ATPγS.** A small dataset of 40,000 particles was picked using a difference of gaussians picker[60], which was used for reference-free 2D classification in Appion[59]. Representative views were selected as templates for template-based particle picking using FindEM[61], which yielded 940,396 putative particle selections. All subsequent processing was performed in RELION 3.1[62]. Particle coordinates were extracted at 3.45 Å/pixel from the motion-corrected/dose-weighted micrographs with a box size of 80 pixels, and 2D classified into 200 classes. Based on this 2D analysis, 564,930 particles belonging to classes displaying details corresponding to secondary structural elements were selected for further processing in 3D (examples shown in Supplementary Fig. 2b). A low-resolution negative stain reconstruction of human LONP1 was used as an initial model for 3D classification of the particles (3 classes, $T = 4$, 25 iterations). One 3D class containing 52% of the particles resembled an open lockwasher (hereafter referred to as substrate-free), while the other two classes resembled the previously determined substrate-bound Lon complex[31] (Supplementary Fig. 2b). The alignment parameters were used to extract 293,886 and 271,044 centered, full-size particles corresponding to the substrate-free and substrate-bound complexes, respectively (1.15 Å/pixel, box size 288 pixels). The substrate-free and substrate-bound 3D classes were scaled to the appropriate pixel size and used as initial models for 3D auto-refinement of these particles, resulting in reconstructions at reported resolutions of ~4.9 and ~4.7 Å according to FSC of half maps at 0.143, respectively. The image shifts imposed to acquire each group of 16 exposures during data collection were used to generate 16 optics groups for CTF refinement[63] (per-particle defocus, per-micrograph astigmatism, and beam-tilt estimation). CTF refinement followed by 3D auto-refinement using a soft-edged 3D mask was repeated three times, ultimately resulting in reconstructions with reported resolutions of ~3.4 and ~3.2 Å for the substrate-free and substrate-bound complexes, respectively (Supplementary Fig. 2b).

Notably, the density for one of the six subunits in the substrate-free reconstruction was almost non-existent, which prompted us to perform a 3D classification without alignment on the particles contributing to this reconstruction (3 classes, $T = 15$, 50 iterations). One of the three classes contained stronger density corresponding to the sixth subunit, and the subset of 45,300 particles contributing to this reconstruction were selected for 3D auto-refinement. The resulting six-subunit structure was determined to have a resolution of ~3.6 Å, although the sixth subunit remained insufficiently resolved for ab initio modeling (colored blue in Supplementary Fig. 2b).

3D classification without alignment was used to identify the particles containing the highest resolution structural data contained in both the substrate-free and

substrate-bound particle stacks. 3D masks encompassing the five subunits of the substrate-free complex, and all six subunits for the substrate-bound complex were used for these classifications (3 classes, $T = 15$, 50 iterations). In total, 83,162 particles from the two highest resolution classes of the substrate-free complex, and 38,130 particles from the highest resolution class of the substrate-bound complex were selected for a final masked 3D auto-refinement using local angular and translational searches (0.9°, 2-pixel search range with a 0.5-pixel step). The resulting reconstructions showed a marginal qualitative improvement in density, although the reported resolutions were unchanged at 3.4 and 3.2 Å for the substrate-free and substrate-bound, respectively (Supplementary Fig. 2c).

**Image processing for WalkerB mutant human LONP1 + ATP.** For WalkerB mutant human LONP1, real-time preprocessing was performed during cryo-EM data collection using the Appion processing environment[59]. Micrograph frames were aligned using MotionCor2[57] and CTF parameters were estimated with CTFFind4[58]. In total, 938,590 particles were selected using a Difference of Gaussian (DoG)-based automated particle picker[60] and were extracted using a box size of 336 pixels. The particle stack was then exported to cryoSPARC v3.0.1[64] for reference-free 2D classification. In total, 229,625 particles were selected from 2D classes and used to create an ab initio reconstruction. The resulting 3D reconstruction, in addition to an ab initio reconstruction created using 100 particles, were used as initial models for 3D heterogenous refinement with three classes. The best class, representing 128,929 of the particles, was used for a final 3D homogeneous refinement. The particles refined to a reported resolution of 4.8 Å as estimated by Fourier Shell Correlation using a cutoff of 0.143.

**Image processing for wild-type human LONP1 + ATPγS+ bortezomib.** Similar to the wild-type human LONP1 + ATPγS dataset, a small dataset of 40,000 particles were picked using a difference of gaussians picker[60], which was used for reference-free 2D classification in Appion[59]. Representative views were selected as templates for template-based particle picking using FindEM[61], which yielded 1,907,098 putative particle selections. All subsequent processing was performed in RELION 3.1[62]. Particle coordinates were extracted at 3.45 Å/pixel from the motion-corrected/dose-weighted micrographs with a box size of 80 pixels, and 2D classified into 200 classes. Based on this 2D analysis, 1,591,575 particles belonging to classes displaying details corresponding to secondary structural elements were selected for further processing in 3D (examples shown in Supplementary Fig. 13b). A low-resolution reconstruction of human LONP1 was used as an initial model for 3D auto-refinement of the particles. The alignment parameters were used to extract centered, full-size particles (1.15 Å/pixel, box size 288 pixels). The unbinned reconstruction possessed a C6-symmetric protease domain and was further processed separately from the asymmetric ATPase domain. CTF refinement followed by 3D auto-refinement, using a soft-edged 3D mask over the protease domains and applying C6 symmetry, was repeated four times, resulting in a symmetric protease domain reconstruction with reported resolution of ~2.9 Å according to FSC of half maps at 0.143 (Supplementary Fig. 13b).

A low-resolution reconstruction of ATPase domains of human LONP1 was used as an initial model for 3D auto-refinement of the 1,591,575 particles previously aligned to the protease domain. The asymmetric ATPase domain was further refined using a soft-edged 3D mask over the ATPases and applying C1 symmetry. Similar to the substrate-bound wild-type LONP1 structure, the density for one of the six subunits in the substrate-free reconstruction was almost non-existent, which prompted us to perform a 3D classification without alignment on the particles contributing to this reconstruction (5 classes, $T = 15$, 50 iterations). One of the five classes contained stronger density corresponding to the sixth subunit, and the subset of 180,627 particles contributing to this reconstruction were selected for 3D auto-refinement. The resulting six-subunit structure was determined to have a resolution of ~3.2 Å by FSC at 0.143, although the sixth subunit remained poorly resolved (colored blue in Supplementary Fig. 13b).

The initial refinement of the ATPase domains was used as a starting model for an asymmetric refinement of the protease domains using a soft-edged 3D mask over the proteases and applying C1 symmetry. 3D classification without alignment on the particles contributing to this reconstruction was performed in order to isolate the highest resolution reconstruction (3 classes, $T = 15$, 50 iterations). One of the three classes consisting of 239,730 particles with the most high-resolution features was selected for 3D auto-refinement. The resulting structure was determined to have a resolution of ~3.0 Å by FSC at 0.143. A final composite map of the asymmetric ATPase and asymmetric protease refinements was generated for atomic model building and refinement using the "vop max" operation in UCSF Chimera[65].

**Atomic model building and refinement.** Model building and refinement were similarly performed for the substrate-bound and bortezomib-bound LONP1 structures. A homology model of human LONP1 was generated using a previously resolved cryo-EM structure of bacterial LONP1 as a starting model using SWISS-MODEL[66]. This initial model was split into ATPase and protease domains and rigid body docked into the density of each of the subunits using UCSF Chimera[65]. Real-space refinement of the docked structures and ab initio model building were performed in COOT[67]. The flexible linker regions were modeled ab

initio and a twelve-amino acid poly-alanine peptide as well as ATP, ADP, magnesium cofactor, and Bortezomib molecules were built into the density corresponding to substrate, nucleotide, and drug, respectively. Further refinement of the full hexameric atomic model was performed using one round each of morphing and simulated annealing in addition to five real-space refinement macrocycles with atomic displacement parameters, secondary structure restraints, local grid searches, non-crystallographic symmetry, Ramachandran restraints, and global minimization in PHENIX[68]. One round of geometry minimization with Ramachandran and rotamer restraints was used to minimize clash scores, followed by a final round of real-space refinement in PHENIX[68].

To create an initial model for substrate-free LONP1, individual subunits from the substrate-bound human LONP1 atomic model were docked into the density of each of the five well-resolved subunits of substrate-free LONP1 using UCSF Chimera[65]. Real-space refinement of the docked structures and ab initio model building were performed in COOT[67]. Residues from the model in were trimmed to Cα unless there was clearly discernible EM density for Cβ. Further refinement of the model was performed using one round each of morphing and simulated annealing in addition to five real-space refinement macrocycles with atomic displacement parameters, secondary structure restraints, local grid searches, non-crystallographic symmetry, Ramachandran restraints, and global minimization in PHENIX[68]. ADP molecules were built into the density corresponding to nucleotide into the five well-resolved subunits. One round of geometry minimization with Ramachandran and rotamer restraints was used to minimize clash scores followed by a final round of real-space refinement in PHENIX[68]. Due to the low resolution of the sixth subunit (blue in Supplementary Fig. 2b) atomic coordinates for this subunit are not included in the deposited atomic model. However, since secondary structural elements were visible for this subunit in the reconstruction, a copy of subunit B from our five-subunit substrate-free model was docked into the density for the sixth subunit for interpretation and figure-making.

UCSF Chimera[65] and ChimeraX[65,69] were used to interpret the EM reconstructions and atomic models, as well as to generate figures.

**Reporting summary**. Further information on research design is available in the Nature Research Reporting Summary linked to this article.

## Data availability

Maps for the substrate-free, substrate-bound, and LONP1[Bz] complexes were deposited in the Electron Microscopy Data Bank (EMDB) under accession IDs EMD-23019, EMD-23020, and EMD-23013, respectively. Corresponding atomic models were deposited in the Protein Data Bank under accession IDs 7KSL, 7KSM, and 7KRZ, respectively. Maps for the Walker B LONP1 complex are available from the EMDB under accession ID EMD-23320. Source data are provided with this paper.

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

## Acknowledgements

We thank J.C. Ducom at Scripps Research High Performance Computing and Charles Bowman at Scripps Research for computational support, as well as Bill Anderson at the Scripps Research Electron Microscopy Facility for microscopy support. We thank Nigel Moriarty for assistance in refinement of the LONP1 bortezomib-bound structure. Further, we thank Carolyn Suzuki (Rutgers) and members of the Lander and Wiseman labs at Scripps Research for insightful and helpful discussions related to this work. M.S. is supported by the National Science Foundation Graduate Research Fellowship Program. G.C.L. is supported by the National Institutes of Health (NIH) AG067594 and an Amgen Young Investigator Award. R.L.W. is supported by the NIH NS095892. G.C.L. and R.L.W. are supported by NIH AG061697. Computational analyses of EM data were performed using shared instrumentation funded by NIH S10OD021634 to G.C.L.

## Author contributions

M.S. prepared the human LONP1 samples, cryo-EM samples, collected data, performed image processing, built the atomic models, performed biochemical experiments, and prepared figures for the manuscript. E.R.W. helped express and purify human LONP1 and build atomic models. A.S.S. collected cryo-EM data for WalkerB mutant LONP1. J.T.M. helped build atomic models. S.J.N. and P.G. performed and analyzed the HDX-MS experiments. M.S., R.L.W., and G.C.L co-wrote the manuscript.

## Competing interests

The authors declare no competing interests.
