## [Peer Review File · Nature Communications]

REVIEWER COMMENTS

Reviewer #1 (Remarks to the Author):

Shin et. al. have determined three cryoEM structures of the Lon protease in human mitochondria: a fully ADP bound structure (albeit obtained in the presence of ATP_rS), an ATP_rS bound with a segment of substrate peptide of unknown source, and a structure with ATP_rS bound and the catalytic active serine covalently modified by bortezomib, a serine protease inhibitor. Comparing these structures with the corresponding ones from the homologous *Y. pestis* Lon leads to the identification of a novel state where substrate binds to the AAA+ domain but the protease active sites exist in an auto-inhibited conformation. The auto-inhibition is relieved by the binding and covalent linkage of bortezomib to the active site.

It is exciting to learn that there is a novel auto-inhibited state where the AAA+ domain is actively engaged in substrate binding whereas the protease domain adopts an inactive conformation. This new state may hold the key to unravel the mechanism by which the Lon peptide translocase is coordinated with its protease activity. Unfortunately, I am concerned that the interpretation of differences between this novel state, i.e. LONP1 closed spiral state (ATP_rS + substrate peptide) and that of *Y. pestis* Lon (Walker B mutant + ATP + substrate peptide) is complicated by the different conditions in which they were obtained. The LONP1 closed spiral structure was captured in the presence of ATP_rS whereas the *Y. pestis* Lon is a walker B mutant and obtained in the presence of ATP. ATP_rS is clearly different from ATP. Therefore, ATP_rS binding may not be able to trigger the kind of allosteric transition allowing the protease active site to adopt the active conformation. To put it another way, the observed auto-inhibited state of LONP1 in the closed spiral conformation may simply be an artifact due to using ATP_rS instead of the physiological ligand ATP. Because the identification of the auto-inhibited state is an important finding of the paper, the authors need to address the issue by, ideally, determining the cryo-EM structure of LONP1 in the same way as *Y. pestis* Lon, i.e. using the Walker B mutation in the presence of ATP, so that we know if ATP (but not ATP_rS) binding is sufficient to activate the protease domain or not.

In addition, I would like also to ask the authors to comment on the unusual observation related to ATP_rS binding to LONP1. Almost 50% of LONP1 particles exist in the open spiral conformation fully occupied by ADP despite being placed in the presence of ATP_rS which can only be very slowly hydrolyzed. Is it because there was too much contaminating ADP in the ATP_rS prep, or that LONP1 was incubated with ATP_rS too long, allowing significant ADP accumulation as a result of ATP_rS hydrolysis? Otherwise, it is hard to imagine that ADP release can be as slow as ATP_rS hydrolysis to result in coexisting open and closed spiral conformations in the presence of saturating ATP_rS.

It is commendable for the authors to use orthogonal methods to validate their cryoEM findings. HDX is a perfect and powerful technique for such purpose. However, there are a few issues with the HDX data analysis/presentation that compromise the validity of the results. Based on the data shown SI Fig 12, the peptide coverage of LONP1 does not appear to be very good. There are quite a few stretches of uncovered regions throughout the sequence of the protein. Moreover, the presentation of the HDX data is confusing: how was the red-blue scale developed? What does the number on the scale mean? For example, does a -12 reading mean 12% less deuterons incorporated? How did the authors condense four HDX time points into a single number? None of these are clearly explained in the M&M section.

I am also worried that the observed HDX difference may not be entirely due to bortezomib induced conformational change. In the M&M section between Line 339 and 340, it reads that 10uM LONP1 was incubated with “the respective ligands” (ATPrS or ATPrS+bortezomib?) at a 1-10 protein-to-ligand molar ratio followed by a 1 to 5 dilution into D2O buffer that did not contain any ligands. This means only 20uM ATPrS was present in the solution of HDX measurements. Is this enough to fully saturate LONP1? In contrast, the concentration of ATPrS used for the cryo-EM sample prep is 1mM. If only 20uM ATPrS was used, a significant amount of LONP1 may be in an apo state and would undoubtedly complicate the interpretation of the observed HDX difference.

I also want to know how the authors dealt with bortezomib modified peptides in their HDX data analysis. It looks like there is some peptide coverage around the active site Ser-855 according to SI Fig 12. How stable is the ester bond between bortezomib and Ser? If bortezomib remains covalently attached to Ser during the workup stage of HDX, there are a number of issues in dealing with the bortezomib modified peptides. There are two exchangeable amide sites on bortezomib. Have they been taken care of when calculating the percentage of deuterium incorporation? The authors assumed some uniform back exchange correction of 70%. Is this justifiable when comparing peptides with or without bortezomib derivative? In this case, although the two peptides may have the same amino acid sequence, one also has bortezomib which may result in differences in elution time, interaction with the HPLC column, etc. These differences would ultimately result in different levels of back exchange. The two exchangeable amide sites on bortezomib can also complicate interpretation of the differential HDX data especially if the peptides covering Ser-855 are short. The HDX difference observed between modified and unmodified peptides may be, at least partly, due to these two amide sites having some level of protection.

Minor points

1. I don't understand the statements made in the discussion regarding substrate selectivity of LONP1 such as those between Line 268 and Line 274. I think the authors need to provide additional clarification on this point. Is there any evidence that LONP1 favors certain types of substrate over the others? If so, how does this kind of selectivity correlate with interactions observed between bortezomib and the LONP1 protease domain? Are there specific interactions formed between bortezomib and LONP1 that suggest certain type of amino acid sequences are favored over others?

2. Because ATPrS is hydrolysable, different incubation times may result in different ADP accumulation levels and relative populations of open to closed spiral states. Therefore, the authors need to specify the ATPrS incubation time when preparing samples for cryo-EM.

3. The authors attribute the presence of substrate to co-purification of endogenous proteins. However this is unlikely if a working purification protocol were used. According to the protocol provided in M&M, no ATP nor ATPrS were included during purification. And because this is the wild-type LONP1 (which is different from the cases cited by the authors where ATP hydrolysis deficient mutants were used), one can safely assume that the LONP1 protein was purified in its apo form with no bound nucleotide (esp. no ATP). It is also well known that only in the presence of slowly/non-hydrolyzable ATP that LONP1 can form stable complex with substrates. Therefore, if the purification procedure worked, very little endogenous protein substrate can survive all the purification steps and co-purified with LONP1. However, prolonged incubation with ATPrS could result in cannibalizing misfolded or partially unfolded LONP1. Therefore, more reason for the authors to control and disclose their incubation time with ATPrS before collecting cryoEM data.

Reviewer #2 (Remarks to the Author):

The strength of this manuscript is the demonstration that cryo EM could be used to solve the oligomeric state of hLon and to detect structural changes induced by nucleotides and/or peptide-based inhibitor. Such approach will significantly benefit the structure and function characterization of hLon as exemplified in the mutation studies to characterize the proposed residues in hLon responsible for substrate translocation.

However, evidences to support the proposed autoinhibited enzyme form and the physiological relevance of the peptide inhibitor released active enzyme form are not compelling. The identity of the peptide substrate mentioned in the autoinhibited complex was never revealed or determined. At one point the authors mentioned the peptide substrate came from an endogenous protein substrate (line 92). What was this protein substrate? What was the evidence that this peptide was a substrate and not a hydrolyzed peptide product? If it was the latter, the “inactive enzyme form” could be a post-catalytic hLon bound the hydrolyzed peptide product. If so, the proposed mechanism will not be valid. According to line 225, there was a 12-residue peptide substrate in the Lon- Bortezomib complex, but according to the legend in Fig 4, the 12- residue polyalanine peptide was modeled into the structure. The rationale for fitting a polyalanine peptide into the proposed unidentified 12-mer peptide substrate was not adequately provided. What was the evidence that a 12 residues polyalanine peptide could be degraded by hLon in the presence of ATP or ATP- γ S? Furthermore, the boronic acid peptide inhibitor is small, its size mimics that of hydrolyzed peptide products generated than a protein substrate containing multiple scissile Lon cleavage sites. Aside from matching the structural symmetry to what has been shown to exit in other Lon homologs, what is the justification to correlate this “inhibitor bound enzyme complex” to be a “released activate hLon form” when additional substrate interaction with other regions of hLon responsible for substrate translocation are missing in Bortezomib. Is ATP hydrolysis required for Bortezomib to form a covalent adduct with hLon? How does Bortezomib fit into the substrate translocation mechanism? An important feature of Lon and other ATP-dependent proteases is the processive protein degradation ability. There was no referral or discussion on how the findings obtained in this study would fit into a processive protein degradation mechanism. The mechanism shown in Fig 5 indicates hydrolyzed peptide products are generated before the protein substrate is completely translocated. This model seems to indicate at least a partially non-processive mechanism. No activity or protein degradation profile data were provided to support this mechanism. Finally, ATP γ S was used to stabilize hLon complex for structural characterization. Since ATP γ S was hydrolyzed by hLon with reduced efficiency, the substrate translocation process would be slowed down accordingly. As such the timing between substrate translocation mediated by the ATPase domain and peptide bond cleavage mediated by the proteolytic site would be different than the ATP mediated reaction. The physiological significance of the enzyme forms generated from ATP γ S is not clear and adequately justified.

Overall this manuscript is strong in structural biology but weak in functional validation. It lacks the appropriate functional and activity data to support the claims deduced based solely on structural characterizations.

Reviewer #3 (Remarks to the Author):

The manuscript by Shin et al., presents several cryo-EM structures of the human mitochondrial LONP1 AAA+ protease. As the authors describe, this is an important target because LONP1 is essential and serves critical roles in mitochondrial homeostasis; additionally, mutations are associated with the CODAS developmental disorder and these structures provide insight into how the mutations disrupt function. The cryo-EM work presented here is of high quality and the manuscript is clear and well-written. Structures of both the substrate-bound and substrate-free complexes are determined, based on two distinct classes identified in the dataset. The substrate-free structure adopts an open, left-handed spiral, similar other 'lockwasher' arrangements of AAA+s, identifying an inactive architecture. The substrate bound complex is in a right-handed spiral with a polypeptide strand trapped in the central channel. This 'spiral staircase' architecture and pore loop-substrate interactions are similar to many other substrate-bound AAA+s. Notably key differences between this structure and one of bacterial LONP1 they published previously are described, including a different arrangement of the spiral seam (one disengaged subunit identified here, compared to two in the bacterial complex), additional substrate contacts involving the secondary pore loops and contact by a Tyr at 599 as well as differences in activation state of the protease. Mutagenesis of Y599 site shows loss of proteolysis activity indicating this residue is indeed important for unfolding. They identify that the protease is in an asymmetric inactive conformation, which they call an "auto-inhibited" state. Importantly they also determine a co-structure with the bortezomib substrate mimic inhibitor showing the protease in an active conformation when substrate is in the pocket. This work provides novel insight about inactive to active conformational changes in the AAA+ and protease rings and overall is an important contribution to the AAA+ protease field and thus publication in Nature Communications. However, a couple concerns are worth noting and should be addressed.

Main concerns:

- 1.) Referring to the wt substrate-bound structure with ATPyS as "auto-inhibited" is misleading since, as they conclude, substrate itself activates the protease (based on the LONP1-BZ structure). What seems to have been captured is simply an inactive conformation. How can an enzyme be inhibited if substrate itself is what provides the binding energy to drive the activation? Typically auto inhibited states are released by the binding of other co-factors or protein components or through modification such that the enzyme does not process substrate in one condition but does in another. Are there conditions in which this LONP1 inhibited state persists in the presence of substrate which can be shown in a functional assay? Is it activated by thermal or oxidative stresses? The function of this state as an 'auto-inhibited' state seems unclear in terms of how this would occur in vivo. It would be better to describe this as an inactive state or to provide additional functional data to indicate where this state fits in the cycle. Nonetheless, as an inactive state this structure is very interesting, revealing that substrate binding to the protease active site is required to drive conformational changes that align the catalytic residues.
- 2.) The authors describe differences between this structure and bacterial one they previously published, indicating these are functionally/evolutionarily significant. Notable differences are in the AAA hexamer and substrate-bound architecture, and that substrate binding to the channel is not sufficient to activate the protease for human LONP1, but is for the bacterial one. However, it is worth noting that these structures were solved under different conditions. The bacterial LONP1 study relied on a Walker B ATPase inactive mutation and utilized ATP as well as a specific substrate, while this study used wildtype enzyme and ATPyS (and an unknown substrate present with the purification) to trap the substrate bound state. Thus, these structural differences could be attributed to differences in the experimental conditions

– the different nucleotides or use of the ATPase inactivating mutant, which could affect conformation and substrate interactions. Thus, it is important for the authors to note these experimental differences and this possibility.

Other minor concerns:

- 1.) Figure 1 is presented in reverse order from how it is described in the text. Showing the substrate free structure in a/b would be preferred, since that is what is discussed first in the text.
- 2.) In Figure 1 it is difficult to tell the conformational differences between substrate free and bound. A cartoon model showing left and right-spirals or alignment of the models would be helpful.
- 3.) Surprisingly, the N-terminal domains were not present in the structure. Although the authors indicate they are flexible, were there any additional classification or refinement procedures performed in an attempt to visualize and improve the resolution of that domain?
- 4.) Figure 1a – move the labels outside the structure as they are difficult to see
- 5.) Figure 1d – include nucleotide state
- 6.) Figure 3. It is unclear what makes the active site in an inhibited state. A comparison with the active site of bacterial LONP1 showing changes in the position of the catalytic residues would be helpful.

Response to Reviewer.

We sincerely thank the reviewers for thoroughly reading our manuscript and providing incisive comments related to our findings. We have taken all the reviewer's comments into careful consideration, and adapted the revised submission to address these comments through inclusion of new structures, figure panels, and adaptations to the text. Below is a point-by-point response to the reviews, describing how each comment was addressed. Our responses are in blue, in-line with the reviewer's original comment.

REVIEWER COMMENTS

Reviewer #1 (Remarks to Author):

Shin et. al. have determined three cryoEM structures of the Lon protease in human mitochondria: a fully ADP bound structure (albeit obtained in the presence of ATP_rS), an ATP_rS bound with a segment of substrate peptide of unknown source, and a structure with ATP_rS bound and the catalytic active serine covalently modified by bortezomib, a serine protease inhibitor. Comparing these structures with the corresponding ones from the homologous Y. pestis Lon leads to the identification of a novel state where substrate binds to the AAA+ domain but the protease active sites exist in an auto-inhibited conformation. The auto-inhibition is relieved by the binding and covalent linkage of bortezomib to the active site.

It is exciting to learn that there is a novel auto-inhibited state where the AAA+ domain is actively engaged in substrate binding whereas the protease domain adopts an inactive conformation. This new state may hold the key to unravel the mechanism by which the Lon peptide translocase is coordinated with its protease activity. Unfortunately, I am concerned that the interpretation of differences between this novel state, i.e. LONP1 closed spiral state (ATP_rS + substrate peptide) and that of Y. pestis Lon (Walker B mutant + ATP + substrate peptide) is complicated by the different conditions in which they were obtained. The LONP1 closed spiral structure was captured in the presence of ATP_rS whereas the Y. pestis Lon is a walker B mutant and obtained in the presence of ATP. ATP_rS is clearly different from ATP. Therefore, ATP_rS binding may not be able to trigger the kind of allosteric transition allowing the protease active site to adopt the active conformation. To put it another way, the observed auto-inhibited state of LONP1 in the closed spiral conformation may simply be an artifact due to using ATP_rS instead of the physiological ligand ATP. Because the identification of the auto-inhibited state is an important finding of the paper, the authors need to address the issue by, ideally, determining the cryo-EM structure of LONP1 in the same way as Y. pestis Lon, i.e. using the Walker B mutation in the presence of ATP, so that we know if ATP (but not ATP_rS) binding is sufficient to activate the protease domain or not.

We thank the reviewer for prompting us to further investigate the possibility that the unique conformation we observe for human LONP1 is a result of using ATP_rS to limit ATP hydrolysis, as opposed to the Walker B mutation used for our previous study of Y. pestis Lon. To address this, we generated a LONP1 construct bearing a Walker B mutation and collected a cryo-EM dataset in the presence of ATP, as we performed previously for Y. pestis LON. At a resolution of ~5 Å, the Walker B LONP1 conformation is indistinguishable from that obtained using ATP_rS. We added **Supplementary Fig. 8** and **Supplementary Fig. 9** to emphasize the similarities between Walker B + ATP and WT+ ATP_rS structures. We feel that this finding strengthens our conclusion that the protease activation mechanism in LONP1 is distinct from that of Y. pestis Lon.

We incorporated the results of this additional study by adding the following sentences into the main text on page 6:

*“To rule out the possibility that this unique configuration of LONP1 is a result of using ATP_rS to slow ATP hydrolysis, as opposed to incorporating a Walker B mutation in the Y. pestis study, we introduced a Walker B mutation into LONP1 (E591 to A591). We determined the structure of this Walker B LONP1 mutant in the presence of ATP, and at a resolution of ~4.8 Å, its conformation is indistinguishable from that of the substrate-bound WT LONP1 in the presence of ATP_rS (**Supplementary Fig. 8**, **Supplementary Fig. 9**, and **Supplementary Table 1**). After extensive 3D classification, all particles belonging to Walker B LONP1 appear to be substrate-bound, despite not adding substrate during protein purification or cryo-EM sample preparation. This suggests that Walker B LONP1 constructs*

form more stable complexes with endogenous co-purified substrate or self-degradation products during purification and sample preparation than WT LONP1.”

In addition, I would like also to ask the authors to comment on the unusual observation related to ATP γ S binding to LONP1. Almost 50% of LONP1 particles exist in the open spiral conformation fully occupied by ADP despite being placed in the presence of ATP γ S which can only be very slowly hydrolyzed. Is it because there was too much contaminating ADP in the ATP γ S prep, or that LONP1 was incubated with ATP γ S too long, allowing significant ADP accumulation as a result of ATP γ S hydrolysis? Otherwise, it is hard to imagine that ADP release can be as slow as ATP γ S hydrolysis to result in coexisting open and closed spiral conformations in the presence of saturating ATP γ S.

We were initially perplexed by this result, but after considering some of the foundational work concerning activation of the Lon holoenzyme, we feel that can be explained by the role of ADP as potent ATP competitive inhibitor. The established mechanism for bacterial and mitochondrial Lon proteases is that substrate binding stimulates nucleotide exchange of ADP for ATP. It was previously shown that in the absence of a proteolytic substrate, ADP has a 10 to 20-fold higher affinity for Lon than ATP (Thomas-Wohlever and Li 2002, Biochemistry), possibly serving as a regulatory mechanism for mitigating futile cycles of ATP hydrolysis *in vivo*. Menon and Goldberg also showed that Lon purified in the absence of nucleotide is bound tightly to ADP and that only the addition of a proteolytic substrate could encourage ADP dissociation (Menon and Goldberg 1987, JBC). This behavior was also shown for LONP1 by Liu et al. (Liu et al. 2004, JBC). Since our LONP1 purification protocol does not include added ATP, the sample will likely have hydrolyzed any available ATP to ADP prior to the 10 minute incubation with ATP γ S and subsequent grid preparation. Because we did not add additional substrate to the sample, there was likely limited substrate available (either co-purified endogenous substrates or self-degradation products) to encourage nucleotide exchange for ADP to ATP γ S in order to produce a stable LONP1-Peptide-ATP γ S complex. Upon checking the purity of our purchased ATP γ S stock, Jena biosciences has listed their product as $\geq 90\%$ ATP with $< 10\%$ ADP, indicating that there is also a significant amount of ADP in the sample, which will compete with the lower affinity ATP γ S association, especially in low proteolytic substrate concentrations. Therefore, we believe that the ADP-bound LONP1 particles in our ATP γ S treated samples are due to low peptide substrate concentrations compounded by contaminating levels of ADP in our ATP γ S stock. In contrast, our LONP1 Walker B mutants were purified similarly but all of the particles are found in the ATP-bound substrate-engaged conformation (described in response to the previous reviewer comment). This complex is likely formed during purification as reduced ATP hydrolysis rates of the Walker B mutant promotes formation of the long-lived, stable LONP1-peptide-ATP complex where the peptide is an unknown endogenous *E. coli* substrate or self-degradation product, as seen for other structural studies of AAA+ proteases (see review from Puchades et al. Nat Rev Mol Cell Biol 2020).

We summarize this rationale for mixed conformational populations in the main text on page 5:

“The presence of a fully ADP-bound LONP1 conformation, despite incubating the sample with saturating amounts of ATP γ S, likely stems from two factors: the LONP1 sample was purified without supplementing ATP, and the presence of contaminating ADP in the ATP γ S stock. Since Lon has a 10-20 fold higher affinity for ADP over ATP,⁴⁴ it is likely that LONP1 was fully ADP-bound prior to incubation with ATP γ S. Since ADP exchange for ATP is promoted by substrate binding,^{45,46} adoption of the substrate-bound state would be limited by the amount of free substrate present in the purified sample.”

It is commendable for the authors to use orthogonal methods to validate their cryoEM findings. HDX is a perfect and powerful technique for such purpose. However, there are a few issues with the HDX data analysis/presentation that compromise the validity of the results. Based on the data shown SI Fig 12, the peptide coverage of LONP1 does not appear to be very good. There are quite a few stretches of uncovered regions throughout the sequence of the protein.

We see limited peptide coverage in N-terminal portion of LONP1 (up to residue 415). However, since this region is not observed in our cryo-EM structure due to its overall mobility relative to the ATPase and protease domains, we have removed this N-terminal portion from **Supplemental figure 16** to more clearly delineate the

residues present within the structure and the apparent coverage. While we hoped to attain 100% coverage within the ATPase and protease regions, we did obtain 88% coverage of the ATPase and protease domains, which included the regions of most specific relevance for our model (i.e., the protease active site). We did perform extensive examination of the HDX data to search for peptides corresponding to the remaining undigested regions; however, it is likely that close inter- or intra-subunit interactions prevented complete digestion of the LONP1 peptides, limiting our ability to obtain complete coverage in these experiments.

Moreover, the presentation of the HDX data is confusing: how was the red-blue scale developed? What does the number on the scale mean? For example, does a -12 reading mean 12% less deuterons incorporated? How did the authors condense four HDX time points into a single number? None of these are clearly explained in the M&M section.

We agree with the reviewer that the HDX data could have been presented more clearly, and have modified these aspects of the manuscript to prevent any confusion. To facilitate 'painting' HDX-MS data onto structures, we first perform a data consolidation where the differences ($\Delta\%D$) from each timepoint are averaged into a single $\Delta\%D$ value. The averaged $\Delta\%D$ values are either represented as a color according to the color scale or colored grey if the average $\Delta\%D$ values are between -5 %D and 5 %D. As an example, a $\Delta\%D$ of 10% from a 20 aa peptide would indicate that there are an average of 2 more amide deuterons found on that peptide. The red-blue scale is a standard way to display HDX data that is accepted by the community, but we agree that it might be confusing to some. In order to address these concerns, we edited the caption of **Supplemental figure 16** to clear up any confusion about the scale:

"D₂O exchange decreased by 0-6% in regions colored blue while regions colored in red depict where the greatest reduction in D₂O exchange occurred, by ~24%."

I am also worried that the observed HDX difference may not be entirely due to bortezomib induced conformational change. In the M&M section between Line 339 and 340, it reads that 10uM LONP1 was incubated with "the respective ligands" (ATPrS or ATPrS+bortezomib?) at a 1-10 protein-to-ligand molar ratio followed by a 1 to 5 dilution into D2O buffer that did not contain any ligands. This means only 20uM ATPrS was present in the solution of HDX measurements. Is this enough to fully saturate LONP1? In contrast, the concentration of ATPrS used for the cryo-EM sample prep is 1mM. If only 20uM ATPrS was used, a significant amount of LONP1 may be in an apo state and would undoubtedly complicate the interpretation of the observed HDX difference.

We realize that we weren't explicit about these details of the HDX experiments. We are confident that these experiments specifically probe the differences upon addition of bortezomib. 1 mM ATPyS was used in all of the HDX experiments and the 1:10 protein-to-ligand molar ratio only pertains to bortezomib. Since bortezomib covalently binds LONP1 with a K_m of 17nM (Lu et al. Mol Cell 2013), we expect that bortezomib will be bound to all subunits of all LONP1 hexamers in the sample prior to addition of D₂O. These points have been clarified in the Materials and Methods on page 16:

"For HDX-MS analysis, LONP1 (10 μ M) was incubated with 1mM TCEP and 1mM ATPyS for 5 minutes on ice and then Bortezomib (100uM) was added and incubated at 37°C for 30 minutes. Next, 5 μ l of sample was diluted into 20 μ l D₂O buffer (50 mM Tris-HCl, pH 8; 75 mM KCl; 10 mM MgCl₂) and incubated for various time points (0, 10, 60, 300, and 900 s) at 4°C. The deuterium exchange was then slowed by mixing with 25 μ l of cold (4°C) 0.1M Sodium Phosphate, 50mM TCEP. Quenched samples were immediately injected into the HDX platform. Upon injection, samples were passed through an immobilized pepsin column (2mm \times 2cm) at 200 μ l min⁻¹ and the digested peptides were captured on a 2mm \times 1cm C₈ trap column (Agilent) and desalted. Peptides were separated across a 2.1mm \times 5cm C₁₈ column (1.9 μ l Hypersil Gold, ThermoFisher) with a linear gradient of 4% - 40% CH₃CN and 0.3% formic acid, over 5 min. Sample handling, protein digestion and peptide separation were conducted at 4°C."

I also want to know how the authors dealt with bortezomib modified peptides in their HDX data analysis. It looks like there is some peptide coverage around the active site Ser-855 according to SI Fig 12. How stable is the ester bond between bortezomib and Ser? If bortezomib remains covalently attached to Ser during the

workup stage of HDX, there are a number of issues in dealing with the bortezomib modified peptides. There are two exchangeable amide sites on bortezomib. Have they been taken care of when calculating the percentage of deuterium incorporation? The authors assumed some uniform back exchange correction of 70%. Is this justifiable when comparing peptides with or without bortezomib derivative? In this case, although the two peptides may have the same amino acid sequence, one also has bortezomib which may result in differences in elution time, interaction with the HPLC column, etc. These differences would ultimately result in different levels of back exchange. The two exchangeable amide sites on bortezomib can also complicate interpretation of the differential HDX data especially if the peptides covering Ser-855 are short. The HDX difference observed between modified and unmodified peptides may be, at least partly, due to these two amide sites having some level of protection.

Bortezomib is a reversible inhibitor of LONP1. Since we observe the presence of peptides including S855 in samples incubated in the presence or absence of bortezomib, this likely reflects effective reversal of covalent modification during the low pH quench and subsequent HPLC fractionation involved in the HDX experiment. However, it is important to note that our HDX analysis demonstrates that the entirety of the protease active site, including two additional peptides that do not contain the active site Ser, also show significant reductions in solvent exposure and thereby, conformational flexibility upon administration of bortezomib. This result is consistent with the remodeling of the protease active site observed in our bortezomib-bound structure and further highlights how engagement of the protease active site is required for effective activation of proteolytic activity.

Minor points

1. I don't understand the statements made in the discussion regarding substrate selectivity of LONP1 such as those between Line 268 and Line 274. I think the authors need to provide additional clarification on this point. Is there any evidence that LONP1 favors certain types of substrate over the others? If so, how does this kind of selectivity correlate with interactions observed between bortezomib and the LONP1 protease domain? Are there specific interactions formed between bortezomib and LONP1 that suggest certain type of amino acid sequences are favored over others?

We agree with the reviewer that our wording here may be an overstatement and could lead to confusion. We have modified the text on page 12 to emphasize that additional regulation of the protease activity could enable a fine-tuning of enzymatic function according to targeted substrates within the mitochondria:

“Notably, this protease activation mechanism would enable human LONP1 to be specifically tuned to the types and quantities of substrates present within the matrix, which continuously fluctuate in response to cellular queues. This fine-tuning of protease activity also presents a mechanism by which LONP1 could perform site-specific cleavage of substrates.⁵⁰ Further, the additional level of regulation of proteolytic activity for LONP1 revealed by our structures provides a mechanism to prevent the aberrant degradation of proteins important for mitochondrial function (Fig. 5).”

2. Because ATP γ S is hydrolysable, different incubation times may result in different ADP accumulation levels and relative populations of open to closed spiral states. Therefore, the authors need to specify the ATP γ S incubation time when preparing samples for cryo-EM.

We have included the incubation times in the revised Materials and Methods section, and mention that the stock of ATP γ S contained contaminating ADP (<10%) on page 17:

“Wildtype human LONP1 was diluted to a concentration of 2.5 mg/ml in 50 mM Tris pH 8, 75 mM KCl, 10 mM MgCl₂, 1 mM TCEP, and 1 mM ATP γ S (Jena Bioscience, purity \geq 90 % (HPLC), contains < 10 % ADP).”

3. The authors attribute the presence of substrate to co-purification of endogenous proteins. However this is unlikely if a working purification protocol were used. According to the protocol provided in M&M, no ATP nor ATP γ S were included during purification. And because this is the wild-type LONP1 (which is different from the cases cited by the authors where ATP hydrolysis deficient mutants were used), one can safely assume that the LONP1 protein was purified in its apo form with no bound nucleotide (esp. no ATP). It is also well known that

only in the presence of slowly/non-hydrolyzable ATP that LONP1 can form stable complex with substrates. Therefore, if the purification procedure worked, very little endogenous protein substrate can survive all the purification steps and co-purified with LONP1. However, prolonged incubation with ATP γ S could result in cannibalizing misfolded or partially unfolded LONP1. Therefore, more reason for the authors to control and disclose their incubation time with ATP γ S before collecting cryoEM data.

We share the reviewer's concerns regarding the presence of the substrate in the ATP γ S dataset. We and other groups studying AAA+ ATPases frequently find substrate trapped within the central channel upon addition of slowly hydrolyzing nucleotide analogs (VAT (Ripstein et al), Cdc48 (Twomey et al), Hsp104 (Gates et al), AFG3L2 (Puchades et al), etc.) as well as Walker B mutant enzymes (YME1 (Puchades et al), Spastin (Sandate et al), ClpXP (Ripstein et al), etc.). Prior mass spec of such samples have not revealed the presence of a specific peptide. Given that we observe the same conformational arrangement of LONP1 when we use ATP γ S or a Walker B mutation, we don't feel that the source of this engaged peptide substantially impacts our conclusions. However, in addition to including the incubation times with ATP γ S in the methods, we also include the possibility that the engaged peptide is a product of cannibalization on page 5:

"The presence of substrate in a subset of the complexes was attributed to either a co-purified endogenous protein substrate or self-degradation product and is consistent with the presence of endogenous substrate observed previously in cryo-EM studies of numerous other AAA+ proteins.^{21,34,35,37,40-43"}

Reviewer #2 (Remarks to the Author):

The strength of this manuscript is the demonstration that cryo EM could be used to solve the oligomeric state of hLon and to detect structural changes induced by nucleotides and/or peptide-based inhibitor. Such approach will significantly benefit the structure and function characterization of hLon as exemplified in the mutation studies to characterize the proposed residues in hLon responsible for substrate translocation.

However, evidences to support the proposed autoinhibited enzyme form and the physiological relevance of the peptide inhibitor released active enzyme form are not compelling. The identity of the peptide substrate mentioned in the autoinhibited complex was never revealed or determined. At one point the authors mentioned the peptide substrate came from an endogenous protein substrate (line 92). What was this protein substrate?

What was the evidence that this peptide was a substrate and not a hydrolyzed peptide product?

The reviewer keenly points out a major issue that the AAA+ community has encountered in its structural characterization of these motors. As discussed in our response to Reviewer 1's minor point 3, an unfolded peptide is often present in the central channel of all cryo-EM structures of hydrolysis-compromised ATPases, and despite efforts to identify these proteins through mass spectrometry and biochemical analyses, the identity of these peptides remains unknown. Even in cases where native substrate is added after purification, it is unknown if the resolved peptide in the channel corresponds to the added substrate. Thus, the ambiguity of the substrate is a longstanding issue in the field, and for this reason we have further softened the language regarding the identity of the polypeptide, including the possibility that the engaged peptide is a product of cannibalization (see above response to Reviewer 1 Minor Comment 3).

However, in an effort to more directly address this reviewer's concerns regarding the possibility that this substrate is a product of hydrolyzed peptide product, we generated a LONP1 construct containing both a Walker B mutation, as well as a mutation in the protease site. We hoped to observe a substrate bound within the central channel, as well as an ordered peptide positioned for cleavage within the protease sites, similar to bortezomib in our LONP1^{Bz} structure. We were previously successful in using this approach to visualize substrate peptide within both the ATPase channel and protease active site of AFG3L2 (Puchades et al. Mol Cell 2019). Unfortunately, our analyses of the dataset revealed a structure that exactly matched the substrate-bound ATP γ S LONP1 structure, with the protease active site in an inactive conformation (see figure below). While disappointing in that we were not able to observe substrate within the proteolytic active site, the structure of the double mutant structure does indicate that the peptide in the central channel does not correspond to a

hydrolyzed peptide product. We are continuing to optimize this mutant construct and sample preparation in pursuit of a LONP1 structure with bound peptide in the protease site, and hope to report our findings in a subsequent manuscript.

LONP1 - Walker B + protease inactive mutant (3.9 Å resolution)

If it was the latter, the “inactive enzyme form” could be a post-catalytic hLon bound the hydrolyzed peptide product. If so, the proposed mechanism will not be valid.

We now realize that we didn't clearly describe our model, and are in complete agreement with the reviewer about the model of protease activation. It is entirely possible that the inactive enzyme conformation of the protease likely represents a post-catalytic hLONP1. The major finding of this study is that the protease domain is regulated differently in human LONP1 vs. bacterial Lon. Our mechanistic proposal stems from our prior observation that peptide binding within the ATPase domains was sufficient to induce a rearrangement of the protease active sites and induce a switch to an activated configuration, whereas in human LONP1 this binding is insufficient for protease activation. In order to increase our confidence in the comparison between the bacterial and human Lon systems, we generated a LONP1 construct bearing a Walker B mutation, as was performed for our prior study of *Y. pestis* Lon, and determined a structure of ATP-bound Walker B LONP1 at a resolution of ~5 Å. We include this structure in our revised manuscript, showing that Walker B LONP1 conformation is indistinguishable from that obtained using ATPrS (**Supplemental Figs. 8 & 9**). These new data strengthen our proposal that protease activation in human LONP1 is distinct from that of *Y. pestis* Lon: While binding of substrate by the ATPase is sufficient to activate the protease domains in *Y. pestis*, it is insufficient for protease activation in human LONP1. As discussed in the revised manuscript, we hypothesize that peptide interaction with the protease site itself induces the switch to the active state, and after hydrolysis and release of the hydrolyzed peptide the protease site returns to the inactive form. We feel that this is an intriguing observation that will be of substantial interest to the field and prompt additional studies exploring the intricacies of human LONP1 protease activation and its diverse functions, as highlighted in our revised discussion.

According to line 225, there was a 12-residue peptide substrate in the Lon- Bortezomib complex, but according to the legend in Fig 4, the 12- residue polyaniline peptide was modeled into the structure. The rationale for fitting a polyaniline peptide into the proposed unidentified 12-mer peptide substrate was not adequately provided. What was the evidence that a 12 residues polyaniline peptide could be degraded by hLon in the presence of ATP or ATP-γS?

The substrate density is likely a result of averaging a variety of different sequences engaged by the ATPases, such that side chain densities become unresolvable. This is commonly observed in cryo-EM structures of AAA+ ATPases. Given the ambiguity of the substrate within the central channel of the ATPases, these densities are typically modeled as polyalanine in the AAA+ field. We clarify this in the revised manuscript on page 7:

*“The ATPases in our human LONP1 reconstruction encircle a density accommodating a 12-residue peptide substrate (modeled as a polyalanine chain due to ambiguity of substrate sequence), which is surprisingly nearly twice the length of the engaged substrate visualized in substrate-bound *Y. pestis* Lon.³¹”*

Furthermore, the boronic acid peptide inhibitor is small, its size mimics that of hydrolyzed peptide products generated than a protein substrate containing multiple scissile Lon cleavage sites. Aside from matching the structural symmetry to what has been shown to exit in other Lon homologs, what is the justification to correlate this “inhibitor bound enzyme complex” to be a “released activate hLon form” when additional substrate interaction with other regions of hLon responsible for substrate translocation are missing in Bortezomib. Is ATP hydrolysis required for Bortezomib to form a covalent adduct with hLon? How does Bortezomib fit into the substrate translocation mechanism?

In the revised manuscript we now clarify our rationale for introducing the bortezomib inhibitor to promote conformational remodeling of the protease active site to the active conformation. Given that substrate binding within the ATPase domains was seemingly detached from protease activation, we were looking for a way to test if the protease domain was able to adopt the activated protease conformation we had previously been observed for bacterial Lon. A prior crystal structure of bortezomib bound to a protease domain of bacterial Lon (PDB:4YPM) matched the activated protease conformation observed in our prior *Y. pestis* Lon, and further showed how a peptide might be positioned for cleavage. Thus we posited that bortezomib might induce the activated conformer in human LONP1. While it would have been much more ideal to observe a full-length peptide within the protease active site, the presence of substrate bound within the ATPase channel, combined with the presence of bortezomib bound to the substrate binding groove of proteolytically active LONP1, provides confirmation that all of the protease sites are able to adopt the active conformation during substrate translocation. As we state in the text, the detachment of protease activation from substrate translocation is a unique feature of hLONP1, and while the bortezomib structure shows that all proteases can accommodate simultaneous activity, it is entirely possible that each protease site functions independently of the others, as discussed in the revised discussion.

In regard to ATP hydrolysis and bortezomib binding, it has been previously shown that addition of bortezomib induces an increase in LONP1 ATPase activity that is on par with the ATPase rate increases observed upon addition of substrate (Lin et al., Structure 2016). This indicates that bortezomib mimics the substrate-induced ATPase activation observed for LONP1. Further, in our analysis of the hLONP1 + ATPγS + bortezomib dataset, we were unable to identify any ADP-bound, substrate-free complexes. Numerous prior studies have shown that the conversion between the ADP-bound, substrate-free structure to the ATP and substrate-bound structure of LONP1 requires both substrate engagement and ATP hydrolysis. Thus, given that the same stock of ATPγS was used for both experiments and our bortezomib reconstruction contains nucleotide density that is clearly consistent with ATP (ATPγS) in the ATPases, the lack of the ADP-bound, substrate-free conformation supports a model whereby bortezomib binding is ATP-dependent or ATP and bortezomib bind cooperatively, and that bortezomib stabilizes the ATPγS-bound conformation.

An important feature of Lon and other ATP-dependent proteases is the processive protein degradation ability. There was no referral or discussion on how the findings obtained in this study would fit into a processive protein degradation mechanism. The mechanism shown in Fig 5 indicates hydrolyzed peptide products are generated before the protein substrate is completely translocated. This model seems to indicate at least a partially non-processive mechanism. No activity or protein degradation profile data were provided to support this mechanism.

This was an oversight on our part that has been addressed in the revised manuscript. Although our data suggest that substrate binding within the ATPase is decoupled from protease activation, processive degradation is still possible. Processivity of degradation arises from continuous translocation via the ATPases, and as substrate is shuttled toward the proteases the substrate interacts with and activates the protease active sites, bind and are hydrolyzed. Since there is no evidence that protein substrates are fully translocated prior to peptide hydrolysis, we posit that translocation occurs concomitantly with hydrolysis. However, this decoupling of ATPase activity and protease activation also provides a mechanism by which to explain the chaperone function of hLONP1, a possibility that we briefly describe in the revised text.

We address this point regarding processivity and chaperone function in the discussion section on page 13:

“Apart from its proteolytic activity, LONP1 also functions as an ATP-dependent chaperone within mitochondria, facilitating the folding of substrates including OXAL1 and NDUFA9.¹¹ Interestingly, protease inactive LONP1 retains chaperone activity, suggesting that chaperoning and proteolytic degradation are distinct, independently regulated functions of this protein.¹¹⁻¹³ Our results indicate that LONP1 can decouple its ATPase and proteolytic activities through the differential engagement of substrates with the AAA+ and protease domains. This suggests that chaperone substrates may have evolved to interact with substrate-bound, proteolytically inactive LONP1, bypassing the allostery associated with protease activation and allowing effective ATP-dependent chaperoning independent of proteolytic activity. While additional studies are required to further pursue this potential mechanism, our results establish a mechanistic framework to define how proteolytic activity and ATPase-dependent unfolding can be differentially regulated, promoting diverse LONP1-dependent remodeling of proteins in mammalian mitochondria.”

Finally, ATPγS was used to stabilize hLon complex for structural characterization. Since ATPγS was hydrolyzed by hLon with reduced efficiency, the substrate translocation process would be slowed down accordingly. As such the timing between substrate translocation mediated by the ATPase domain and peptide bond cleavage mediated by the proteolytic site would be different than the ATP mediated reaction. The physiological significance of the enzyme forms generated from ATPγS is not clear and adequately justified.

We agree with the reviewer's assessment of the shortcomings of an ATP hydrolysis-deficient construct, and are currently making efforts to visualize an actively translocating wild type LONP1 complex as a means of providing more physiologically relevant mechanistic insights. However, we feel that the data presented, now with the inclusion of the Walker B mutation, provides conclusive evidence of our major finding – that while substrate-binding within the ATPases and protease domains are inextricably linked in bacterial Lon, these are decoupled in hLONP1.

Reviewer #3 (Remarks to the Author):

The manuscript by Shin et al., presents several cryo-EM structures of the human mitochondrial LONP1 AAA+ protease. As the authors describe, this is an important target because LONP1 is essential and serves critical roles in mitochondrial homeostasis; additionally, mutations are associated with the CODAS developmental disorder and these structures provide insight into how the mutations disrupt function. The cryo-EM work presented here is of high quality and the manuscript is clear and well-written. Structures of both the substrate-bound and substrate-free complexes are determined, based on two distinct classes identified in the dataset. The substrate-free structure adopts an open, left-handed spiral, similar other 'lockwasher' arrangements of AAA+s, identifying an inactive architecture. The substrate bound complex is in a right-handed spiral with a polypeptide strand trapped in the central channel. This 'spiral staircase' architecture and pore loop-substrate interactions are similar to many other substrate-bound AAA+s. Notably key differences between this structure and one of bacterial LONP1 they published previously are described, including a different arrangement of the spiral seam (one disengaged subunit identified here, compared to two in the bacterial complex), additional substrate contacts involving the secondary pore loops and contact by a Tyr at 599 as well as differences in activation state of the protease. Mutagenesis of Y599 site shows loss of proteolysis activity indicating this residue is indeed important for unfolding. They identify that the protease is in an asymmetric inactive conformation, which they call an "auto-inhibited" state. Importantly they also determine a co-structure with the

bortezomib substrate mimic inhibitor showing the protease in an active conformation when substrate is in the pocket. This work provides novel insight about inactive to active conformational changes in the AAA+ and protease rings and overall is an important contribution to the AAA+ protease field and thus publication in Nature Communications. However, a couple concerns are worth noting and should be addressed.

Main concerns:

1.) Referring to the wt substrate-bound structure with ATP γ S as “auto-inhibited” is misleading since, as they conclude, substrate itself activates the protease (based on the LONP1-BZ structure). What seems to have been captured is simply an inactive conformation. How can an enzyme be inhibited if substrate itself is what provides the binding energy to drive the activation? Typically auto inhibited states are released by the binding of other co-factors or protein components or through modification such that the enzyme does not process substrate in one condition but does in another. Are there conditions in which this LONP1 inhibited state persists in the presence of substrate which can be shown in a functional assay? Is it activated by thermal or oxidative stresses? The function of this state as an ‘auto-inhibited’ state seems unclear in terms of how this would occur in vivo. It would be better to describe this as an inactive state or to provide additional functional data to indicate where this state fits in the cycle. Nonetheless, as an inactive state this structure is very interesting, revealing that substrate binding to the protease active site is required to drive conformational changes that align the catalytic residues.

We thank the reviewer for pointing this out, and agree that what we observe is not “auto-inhibited,” and rather an inactive conformation. We have replaced this nomenclature throughout the text in the revised manuscript, referring to the protease active site as either “active” or “inactive”.

2.) The authors describe differences between this structure and bacterial one they previously published, indicating these are functionally/evolutionarily significant. Notable differences are in the AAA hexamer and substrate-bound architecture, and that substrate binding to the channel is not sufficient to activate the protease for human LONP1, but is for the bacterial one. However, it is worth noting that these structures were solved under different conditions. The bacterial LONP1 study relied on a Walker B ATPase inactive mutation and utilized ATP as well as a specific substrate, while this study used wildtype enzyme and ATP γ S (and an unknown substrate present with the purification) to trap the substrate bound state. Thus, these structural differences could be attributed to differences in the experimental conditions – the different nucleotides or use of the ATPase inactivating mutant, which could affect conformation and substrate interactions. Thus, it is important for the authors to note these experimental differences and this possibility.

This is an important concern that we addressed by determining a structure of LONP1 bearing a Walker B mutation. Please see our response to Reviewer 1, who raised the same issue (copied below):

We thank the reviewer for prompting us to further investigate the possibility that the unique conformation we observe for human LONP1 is a result of using ATP γ S to limit ATP hydrolysis, as opposed to the Walker B mutation used for our previous study of *Y. pestis* Lon. To address this, we generated a LONP1 construct bearing a Walker B mutation and collected a cryo-EM dataset in the presence of ATP, as we performed previously for *Y. pestis* LON. At a resolution of ~5 Å, the Walker B LONP1 conformation is indistinguishable from that obtained using ATP γ S. We added **Supplementary Fig. 8** and **Supplementary Fig. 9** to emphasize the similarities between Walker B + ATP and WT+ ATP γ S structures. We feel that this finding strengthens our conclusion that the protease activation mechanism in LONP1 is distinct from that of *Y. pestis* Lon.

Other minor concerns:

1.) Figure 1 is presented in reverse order from how it is described in the text. Showing the substrate free structure in a/b would be preferred, since that is what is discussed first in the text.

We changed the order of the panels to match the text.

2.) In Figure 1 it is difficult to tell the conformational differences between substrate free and bound. A cartoon model showing left and right-spirals or alignment of the models would be helpful.

We revised Figure 1 to include the same views of the hLONP1 substrate-free and substrate-bound structures, which we hope emphasizes the difference in quaternary organization. The change in handedness associated with substrate binding is further emphasized in the model figures shown in Figure 5.

3.) *Surprisingly, the N-terminal domains were not present in the structure. Although the authors indicate they are flexible, were there any additional classification or refinement procedures performed in an attempt to visualize and improve the resolution of that domain?*

We indeed made an effort to resolve a 3D structure of the N-terminal domain of the complex, but were stymied by flexibility of the domains. The revised manuscript contains 2D averages showing the N-terminal domains visualized as fuzzy clouds (**Supplementary Figure 3**).

4.) *Figure 1a – move the labels outside the structure as they are difficult to see*

This has been corrected in the revised text.

5.) *Figure 1d – include nucleotide state*

This has been corrected in the revised text.

6.) *Figure 3. It is unclear what makes the active site in an inhibited state. A comparison with the active site of bacterial LONP1 showing changes in the position of the catalytic residues would be helpful.*

We have revised Figure 3 to include an overlay of all the protease domains from the substrate-bound hLONP1 structure above an overlay of all the protease domains from the substrate-bound *Y. pestis* Lon structure. This emphasizes that despite substrate being present in the ATPase channel of both of these structures, the protease domains of hLONP1 remain inactive, whereas the *Y. pestis* Lon domains are active.

REVIEWER COMMENTS

Reviewer #1 (Remarks to the Author):

I am satisfied with the authors' response and changes made in the manuscript. I think all concerns from the reviews have been adequately addressed.

Reviewer #2 (Remarks to the Author):

The revised manuscript is acceptable.

Reviewer #3 (Remarks to the Author):

The revised manuscript from Shin et. al., is improved and the authors have appropriately addressed reviewer concerns. It is suitable for publication in Nature Communications and will be an important contribution to the AAA+ field. One minor suggestion is that the ATPase domain figure labels in Figure 1C/D should be moved or duplicated for Figure 1 A/B, since the substrate free structure is now the first figure shown.

We are pleased that our revised manuscript satisfied the reviewers' concerns. The only remaining critique to address is from Review 3:

“Reviewer #3 (Remarks to the Author):

The revised manuscript from Shin et. al., is improved and the authors have appropriately addressed reviewer concerns. It is suitable for publication in Nature Communications and will be an important contribution to the AAA+ field. One minor suggestion is that the ATPase domain figure labels in Figure 1C/D should be moved or duplicated for Figure 1 A/B, since the substrate free structure is now the first figure shown.”

Response:

We moved the NTD3H label to panel (A) but feel it's unnecessary to label the subunits for the substrate-free state, since 1) All subunits are bound to ADP, so there is no need to differentiate between nucleotide states of the subunits, and 2) Individual subunits of the substrate-free conformer are never referred to by name in the manuscript. Labeling the subunits with the same labels as the substrate-bound structure in (B) would add further confusion, since none of the subunits are bound to ATP.